# The secondary somatosensory cortex gates mechanical and heat sensitivity

Daniel G. Taub [1,2], Qiufen Jiang[1,2], Francesca Pietrafesa[1,2], Junfeng Su [1,2], Aloe Carroll [3], Caitlin Greene[1,2], Michael R. Blanchard[2], Aakanksha Jain[1,2], Mahmoud El-Rifai[2], Alexis Callen[4], Katherine Yager[4], Clara Chung[5], Zhigang He [1,2], Chinfei Chen [1,2] & Clifford J. Woolf [1,2] ✉

The cerebral cortex is vital for the processing and perception of sensory stimuli. In the somatosensory axis, information is received primarily by two distinct regions, the primary (S1) and secondary (S2) somatosensory cortices. Top-down circuits stemming from S1 can modulate mechanical and cooling but not heat stimuli such that circuit inhibition causes blunted perception. This suggests that responsiveness to particular somatosensory stimuli occurs in a modality specific fashion and we sought to determine additional cortical substrates. In this work, we identify in a mouse model that inhibition of S2 output increases mechanical and heat, but not cooling sensitivity, in contrast to S1. Combining 2-photon anatomical reconstruction with chemogenetic inhibition of specific S2 circuits, we discover that S2 projections to the secondary motor cortex (M2) govern mechanical and heat sensitivity without affecting motor performance or anxiety. Taken together, we show that S2 is an essential cortical structure that governs mechanical and heat sensitivity.

Top-down control of somatosensory encoding by the brain allows for context-dependent modulation of behavioral responses to sensory stimuli based on changing intrinsic or environmental conditions[1,2]. Somatosensory behaviors rely on this top-down control to accurately predict, evaluate, and appropriately react to mechanical, thermal, and chemical stimuli[2,3]. We and others have found that the primary somatosensory cortex (S1) controls somatosensory reflexive behaviors through excitatory corticospinal neurons that innervate the dorsal horn of the spinal cord[4,5]. This constitutes a loop whereby incoming sensory information ascends to S1 for processing and through descending excitatory projections, subsequent information flow in the spinal cord is amplified to facilitate accurate behavioral responses. Inhibition of this S1 corticospinal circuit, accordingly, produces decreased mechanical sensitivity[4]. However, S1 does not encode all somatosensory modalities, with a strong bias toward mechanical and cooling inputs and a notable absence of heat encoding[4,6,7]. This suggests that distinct anatomical regions may participate in the full spectrum of somatosensory control. We sought to identify other cortical regions that could encode the properties that S1 does not. The secondary somatosensory cortex (S2) is an adjacent cortical region that also processes somatosensory stimuli[8–10]. It has been proposed, based on cortico-cortical and thalamo-cortical neural circuitry, latency of responses, and homology to other sensory cortical areas, that S1 and S2 exist in a hierarchy, with S2 as the higher-order cortical area, processing distinct features of the somatosensory experience similar to the visual system[10–12]. The mouse whisker system has provided some evidence supporting this model and subsets of neurons in S2 appear to encode behavioral choice and recalling of past experiences[10,13,14]. However, whether this is the case for somatic stimuli from the body is unclear. S1 also encodes complex aspects of the somatosensory experience questioning the postulated hierarchical relationship and favoring instead the processing of information in parallel[15,16]. Evidence for parallel information processing, where differential modalities are processed within S1 and S2 is accumulating[17], particularly in humans[18,19].

[1]F. M. Kirby Neurobiology Center and Department of Neurology, Boston Children's Hospital, Boston, MA, USA. [2]Department of Neurobiology, Harvard Medical School, Boston, MA, USA. [3]College of Sciences, Northeastern University, Boston, MA, USA. [4]Morrissey College of Arts and Sciences, Boston College, Chestnut Hill, MA, USA. [5]Department of Neuroscience, Boston University, Boston, MA, USA. ✉e-mail: Clifford.Woolf@childrens.harvard.edu

We hypothesized that the secondary somatosensory cortex (S2) may be a crucial substrate for those somatosensory modalities that S1 does not encode. Using a combination of optogenetics and chemogenetics, we now identify the hindpaw area of S2 as a cortical region whose output, in contrast to S1, mitigates mechanical and heat sensitivity. Circuit mapping studies combined with intersectional circuit manipulation strategies identify the secondary motor cortex (M2) as the target of S2 that governs somatosensory behavioral sensitivity. These findings reveal that S2 is a key controller of evoked somatosensory behaviors in a manner quite distinct from S1 circuits and one that is dependent on cortico-cortical connectivity to suppress specific somatosensory responses.

## Results

### Secondary somatosensory cortex inhibition enhances sensitivity

To determine if the secondary somatosensory cortex (S2) has a role in somatosensory behaviors we targeted expression of channelrhodopsin (ChR2) or mCherry into parvalbumin (PV) inhibitory interneurons in the hindlimb region of mouse S2 virally to induce inhibition of the output from S2 (Fig. 1a–c; Supplementary Fig. 1)[20,21]. Our viral injection strategy targeted S2 by injecting through the barrel cortex (part of the whisker trigeminal system and independent from the spinal ascending tract) to avoid any bleed-through into the laterally adjacent insular cortex which is important for thermosensory perception[6] (Fig. 1a). Implantation of an optical fiber to deliver 40 Hz blue light induced strong regional inhibition of the S2 output pyramidal neurons by the activation of PV inhibitory interneurons. Slice recordings from the S2 region confirmed that PV inhibitory interneurons virally transduced with ChR2 are readily excited by blue light and in consequence, adjacent pyramidal neuron activity is suppressed (see also refs. [20,21]) (Fig. 1d, e). Further supporting this, strong suppression of pyramidal neuron expression of the immediate early gene cFos, a proxy of neural activity, was achieved in ChR2 animals exposed to blue light in vivo (Fig. 1f and Supplementary Fig. 1). As there was some limited viral bleedthrough into the adjacent S1 barrel cortex, we also confirmed in control experiments that virus injection and fiber placement into the S1 barrel cortex had no effect on hindpaw somatosensory behaviors (Supplementary Fig. 2).

We then assayed mechanical sensitivity in both the hindpaw contralateral (virus affected) and ipsilateral (unaffected) to the cortical injection site, using graded mechanical von Frey filaments. Stimulation of PV-ChR2 animals with blue light increased mechanical sensitivity compared to PV-mCherry controls (mean: 0.8594 g ± 0.137 mCherry vs. 0.2659 g ± 0.079 ChR2) (Fig. 1g). Only the contralateral paw was affected, confirming that there is no effect on the non-injected hemisphere and that this effect is intratelecephalic (mean: 0.6799 g ± 0.119 mCherry vs. 0.7911 g ± 0.103 ChR2) (Fig. 1g). To ascertain whether the effect on tactile sensitivity was due to the production of allodynia (low-threshold stimuli now being perceived as noxious) vs. non-specific hypersensitivity, we tested a range of low-threshold and high-threshold mechanical stimuli. PV-ChR2 animals, but not PV-mCherry controls, displayed clear allodynia with enhanced responses only to low threshold stimuli (0.04–0.4 g) but not higher intensity mechanical stimuli (0.6–1.4 g) (Fig. 1h).

To assess whether the enhanced sensitivity to mechanical stimuli resulting from the inhibition of S2 extends to other somatosensory modalities, we examined cold and heat sensitivity using either an application of the evaporative coolant acetone and examining the duration of hindpaw withdrawal behaviors or a heat ramp applied until hindpaw withdrawal was observed (Hargreaves' Test[22]). We did not find any behavioral differences in cold sensitivity between PV-mCherry and PV-ChR2 animals (mean: 2.34 s ± 0.674 mCherry vs. 2.67 s ± 0.740 ChR2) (Fig. 1i). However, blue light exposure in PV-ChR2 animals but not mCherry control animals elicited a significantly decreased

response latency on the contralateral but not the ipsilateral paw, indicating enhanced heat sensitivity (mean: 11.50 s ± 1.504 mCherry vs. 5.903 s ± 0.656 ChR2) (Fig. 1j). Interpolating the temperature at which the animals produce a withdrawal response yielded an average temperature of 39 °C in PV-ChR2 animals compared to 44 °C in mCherry controls (Fig. 1k). Therefore, inhibition of hindlimb S2 by activation of PV inhibitory interneurons produces an increase in tactile and heat sensitivity, suggesting that outputs from S2 gate these sensory responses in a manner distinct from S1 which governs tactile and cold sensory responses.

Based on the neural connectivity of rodent and primate S2 and from activity recordings in S2, it might be expected that this cortical region is actively involved in the processing of the aversive components of somatosensory behavior[23–26]. We tested whether S2 output inhibition by PV inhibitory interneuronal activation results in a change in aversive behavior, using a conditioned place aversion assay (Fig. 1l). Mice were conditioned to associate optogenetic inhibition of S2 output with a certain colored chamber (striped vs. solid walls). The floors of both chambers were set at 39 °C, the interpolated average withdrawal-evoking temperature in ChR2-stimulated mice (Fig. 1k). However, after associative training, we observed no aversive response to S2 output inhibition, with mice preferring both chambers equally (Fig. 1m, n) (% time spent in conditioned chamber: mCherry pre-conditioned 51.681 ± 5.156% vs. post-conditioned 49.620 ± 4.794% and ChR2 pre-conditioned 48.751 ± 4.142% vs. post-conditioned 47.275 ± 4.188%). This suggests that while inhibition of S2 increases sensitivity to mechanical and heat evoked behaviors it is not driving an aversive component.

To rule out that S2 induces an anxiolytic-like behavior which could alter somatosensory reactivity, we used an elevated plus maze assay, in which the preference of mice to open or closed arms of an elevated platform are evaluated. This revealed that S2 inhibition did not induce any change in preference to the closed arms compared to mCherry controls (% time spent in open arms mCherry 10.30% ± 4.047 vs 4.38% ± 2.729 ChR2) (Supplementary Fig. 3). This suggests that S2 is primarily involved in the modulation of behavioral responses to somatosensory inputs but not in an affective or aversive fashion.

### PV neurons in S2 preferentially respond to high threshold stimuli

As inhibition of S2 pyramidal neuron output by activation of parvalbumin (PV) inhibitory interneurons increases somatosensory thresholds, we hypothesized that they may function as a gate that governs the behavioral responses to mechanical and thermal somatosensory stimuli. To test this, we used in vivo calcium imaging in awake mice by virally targeting the S2 PV inhibitory interneurons with the calcium indicator GCaMP6f and performing fiber photometry (Fig. 2a, b). Implanted mice had normal somatosensory mechanical detection thresholds, with about 40% of the mice responding to stimulation with a 0.6 g and 60% to a 1.4 g mechanical force (Fig. 2c). Analyzing those trials in which mice displayed a behavioral withdrawal response revealed that in these cases, S2 PV neurons respond with robust transients 1.3–1.5 s after 0.16 g through 1.4 g mechanical force application with time 0 reflecting the first contact of the filament with the paw (Fig. 2d–g). These temporal dynamics are in line with other fiber recordings made during paw stimulation using GCaMP6f[27–30]. In the trials in which the mouse did not withdraw its paw within the 0.04–0.16 g range, had in contrast, no calcium transient response, indicating that PV activity correlates with the behavioral response (Fig. 2d–g). However, once in the noxious range of 1.4 g, calcium responses in the S2 PV neurons occurred even in the absence of a paw withdrawal, indicating that S2 activity and behavioral response are not always coupled once high-intensity stimuli are perceived (Fig. 2g). This suggests circuitry from S2 are able to suppress reflexive behavioral responses to certain degrees of stimulus intensity (largely gating low threshold mechanical stimuli) before spinal withdrawal reflexes

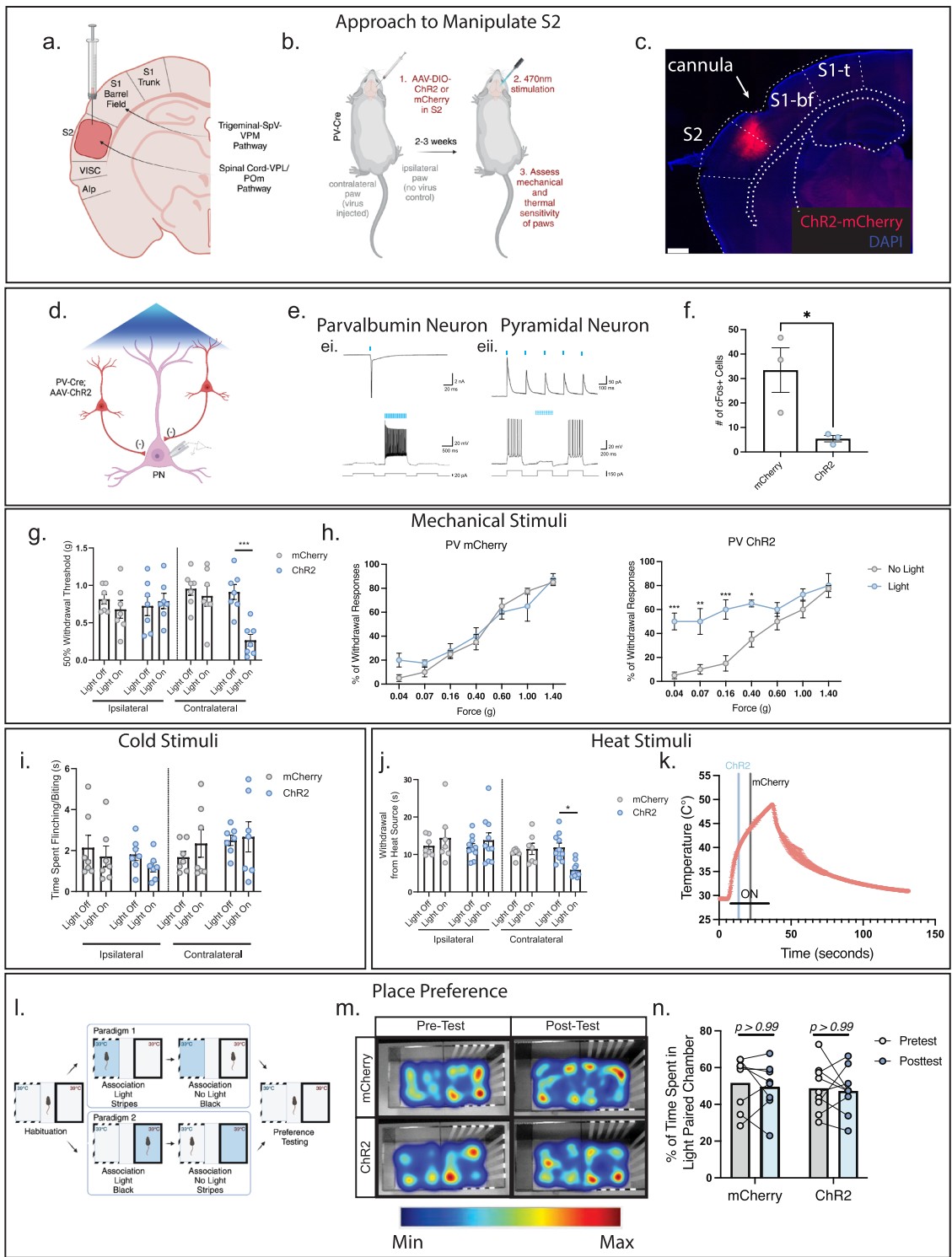

become dominant. Interestingly, the variance in the peak time of individual trials also decreased with higher force stimulation (Fig. 2h), with 1.4 g producing the most temporally coherent response.

To confirm the calcium transients we measure are due to sensory input (stimulation to the paw) rather than motor output (hindpaw withdrawal), we utilized an approach similar to Musall and colleagues[31] and examined calcium transients when the mouse made a task-independent hindpaw withdrawal. In contrast to the time-dependent calcium response we observed during 0.16–1.4 g force stimulations, hindpaw withdrawals in the absence of sensory stimulation produced no apparent calcium transient (Supplementary Fig. 4a). Taken together, this suggests that PV interneurons in S2 respond to sensory

information, they are more responsive to higher mechanical forces on the paw, and fire more synchronously at these forces.

Examination of the calcium transients elicited by PV interneurons during a heat ramp stimulus applied until hindpaw withdrawal (see Methods), also showed a significant increase in activity beginning ~0.72 s prior to the paw withdrawal event and peaked ~1.08 s post paw withdrawal (Fig. 2i). The amplitude of the calcium response to heat was consistently higher than that for mechanical stimuli suggesting a potential preference of PV+ neural recruitment in S2 to noxious heat stimuli (Fig. 2i). In addition, the calcium transients in S2 PV neurons increased almost a full second prior to the motor withdrawal response being initiated, further indicating that the activity recorded in these

**Fig. 1 | Optogenetic inhibition of the secondary somatosensory cortex (S2) enhances tactile and heat sensitivity. a** Injection strategy into S2. **b** Optogenetic inhibition of S2. Cre-dependent channelrhodopsin (ChR2) was injected into the S2 region of parvalbumin (PV)-Cre animals and an optical fiber placed above. **c** ChR2 virus expression in S2 PV neurons. Scale bar: 500 μm. See Supplementary Fig. 1. **d** Diagram depicting slice electrophysiology approach. S2 pyramidal neurons were recorded from while PV-interneurons were activated by blue light. **e** Light-evoked responses in a ChR2$^+$ PV interneuron (**ei**) and a ChR2$^-$ pyramidal neuron (**eii**). **ei** Blue light triggers action potential firing in the PV interneuron in both voltage (top panel) and current (bottom panel) clamp modes. Example trace from 4 PV neurons from 4 animals. **eii** Photoactivation of PV interneurons evokes inhibitory post-synaptic currents (top panel) and suppresses action potential firing (bottom panel) in a pyramidal neuron. L5 example neuron shown from 10 cells/4 animals. Similar inhibition observed in L2/3 neurons. **f** PV neuron activation with blue light in S2 suppresses c-fos expression in pyramidal neurons. Two-tailed unpaired t-test ($n = 3$ per group $p = 0.0382$). **g** Optogenetic inhibition of S2 produces increased

mechanical sensitivity by von Frey. Two Way ANOVA with Tukey's ($n = 7$ per group ChR2 contralateral, light on vs. off $p = 0.0017$). **h** Optogenetic inhibition of S2 produces an allodynic-like state. Two Way ANOVA with Sidak's ($n = 4$ per group) For PV-ChR2 light vs. no light ($0.04$ g $- p = 0.0001$, $0.07$ g $- p = 0.0008$, $0.16$ g $- p = 0.001$, $0.4$ g $- p = 0.0178$). **i** Optogenetic inhibition of S2 does not affect cold sensitivity. Two Way ANOVA with Tukey's ($n = 7$ per group). **j** Optogenetic inhibition of S2 produces increased sensitivity to a heating ramp stimulus. Two Way ANOVA with Tukey's ($n = 7$ mCherry, $n = 10$ ChR2). **k** Thermal profile of the heat ramp stimulus with approximate withdrawal temperatures for both mCherry and ChR2 animals stimulated with blue light depicted (profile from one mouse, three trials). **l** Depiction of the conditioned place preference test and timeline. **m** Inhibition of S2 does not produce conditioned aversive behavior. **n** Time spent in the paired chamber does not differ between ChR2 and mCherry animals. Two Way ANOVA with Sidak's ($n = 8$ mCherry, $n = 9$ ChR2). Data as mean ± SEM. $^*p = <0.05$, $^{**}p = <0.005$, $^{***}p = <0.0005$. See Source Data. Illustrations generated with Biorender.com.

inhibitory interneurons is due to sensory input rather than a response to motor output (Fig. 2i).

We also examined the responsivity of S2 to cold stimuli by applying the evaporative coolant acetone and found that S2 is responsive to cold with similar dynamics to that of a 0.6 g mechanical stimuli, both in amplitude and timing, with transients peaking ~1.5 seconds following the stimulus application but are much less responsive than to a heat stimulus (Fig. 2j). These results indicate that S2 PV interneurons are responsive to a broad set of mechanical and thermal somatosensory stimuli and are particularly responsive to high intensity mechanical and heat stimuli.

We then used a fiber photometry approach to examine excitatory neuron responsivity in S2 using GCaMP6f driven by the CaMKII promoter to compare the dynamics between inhibitory and excitatory populations. Interestingly, we found that populations of excitatory neurons display small and heterogeneous signals to mechanical stimulation. Specifically, a robust transient was only elicited in one condition (von Frey 0.6 g) regardless of the behavioral outcome (Supplementary Fig. 4d–g). This is likely due to the heterogenous, mixed responsivity of excitatory neuron populations to mechanical stimuli, leading to smaller photometry signals, which is similar in nature and organization to that observed in vibrotactile encoding in S1 and the visual cortex[21,32,33]. In contrast, heat stimuli elicited more robust and consistent transients (Supplementary Fig. 4l, m). In comparison with PV neuron activity, PV population activity during a heat ramp peaks prior to CaMKII excitatory neurons in line with PV neurons being fast-spiking and able to gate incoming information. Together with the results that increasing PV activity produces mechanical and heat behavioral hypersensitivity, these data suggest that S2 acts as a controller of behavioral reactivity to both mechanical and heat stimuli.

## PV neuron responsivity in S2 shifts during peripheral inflammation

Based on these optogenetic and fiber photometry data in normal states, we reasoned that increasing peripheral inputs in the setting of peripheral inflammation could shift the responsivity of S2 PV neurons and modify their gating properties. To assess this, we performed fiber photometry during the peak phase of an inflammatory insult to the hindpaw in which the fungal ligand zymosan was injected[34]. Inflammation induced by zymosan significantly shifted the mechanical sensitivity of PV interneurons to a more hypersensitive state (Fig. 2k–n) with the neurons responding to lower mechanical forces than at baseline. In contrast to the basal state, low intensity 0.04 g stimuli now elicited a calcium response in S2 PV neurons and the 0.16 g response was enhanced (Fig. 2k, l). Further, the threshold at which behavioral and S2 calcium responses become uncoupled (in which we hypothesize spinal reflexes become dominant) also shifted, and 0.6 g stimuli now elicited a response in S2 inhibitory interneurons regardless of the

behavioral response (Fig. 2m). Additionally, the time to rise of the transients became more temporally coherent for the 0.16 g stimulus, suggesting that PV neurons now alter their tuning properties to respond to lower threshold stimuli (Fig. 2h). In response to a heat ramp stimulus, the calcium transient from S2 PV neurons was shifted slightly temporally indicating that the temporal dynamics of this population are also altered in response to noxious heat (Fig. 2i, right panel).

Examination of CaMKII excitatory neuron activity was again variable (Supplementary Fig. 4h–k). However, it was noteworthy that the calcium transient observed at 0.6 g at baseline was now absent, concomitant with the increase in PV neuron activity during inflammation (Supplementary Fig. 4f, j). In contrast to PV neurons, in response to a heat ramp stimulus, CaMKII neuron responsivity during inflammation remained relatively unchanged but the slight temporal shift as observed with PV neurons is still evident (Supplementary Fig. 4l, m). This suggests that complex changes occur to both populations during inflammation and that under these conditions, the excitatory/inhibitory dynamics in S2 are altered which may contribute to the robust behavioral hypersensitivity.

## S2 projects to the secondary motor cortex (M2)

Unlike the primary somatosensory cortex (S1), S2 does not project to the dorsal horn of the lumbar region of the spinal cord where hindpaw sensation and movement is generated[5,35]. We hypothesized that a novel circuit must be responsible for altering the stimulus-evoked somatosensory behavior. To define this circuit, we injected an AAV encoding tdTomato under the CAG promoter across all layers of the S2 cortex and performed serial 2-photon tomography[36]. We then aligned these images with the Allen Brain Atlas common framework[37] to identify local and long-distance projection targets (Fig. 3a – quantified in Fig. 3g). We found that S2 makes significant cortical and subcortical projections, the densest of which were observed subcortically in the ventrolateral (VPL) and posterior complexes (Po) of the thalamus, where limb sensory information is received from the spinal cord[38]. These are likely both feedback projections and part of the cortico-thalamo-cortical loop circuitry[12,38]. Other subcortical projection targets included the superior colliculus (SCo), the caudate putamen (CPu) (shown in Fig. 3e, f (arrow)), the periaqueductal gray (PAG), and cervical corticospinal tract, all of which have been implicated in somato-motor circuitry[39]. S2 also made significant corticocortical projections, the densest of which were local to the adjacent auditory/temporal association cortices (AUD/TEa). As observed by others, S2 made significant connections to the contralateral S2, as well as ipsilateral/contralateral projections to the primary somatosensory cortex (S1) (Fig. 3a, f) and to the ipsilateral primary motor cortex (M1)[40] (Fig. 3e). We also observed a major projection to the ipsilateral prefrontal cortex, specifically the secondary motor cortex (M2)[41] (Fig. 3a (orange arrow) Fig. 3c, d). These projection targets place S2 within the common

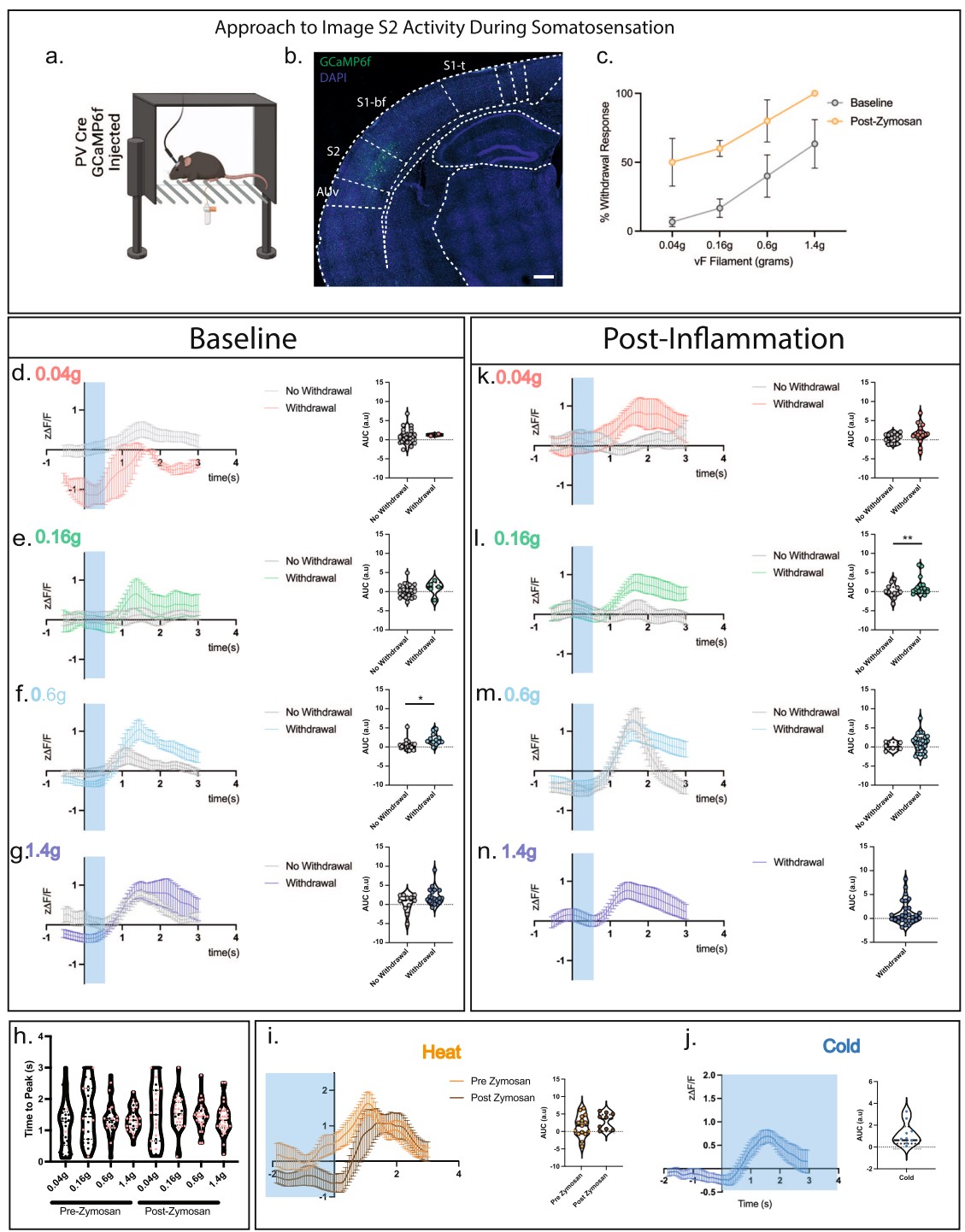

somatomotor architecture[42], but also reveal engagement with diverse neural substrates capable of transducing and calculating sensory information.

The secondary motor cortex (M2) receives sensory inputs and exerts complex effects on behavior. Indeed, during a sensory decision-making task, widefield imaging demonstrates sensory to frontal waves of activity that occurs before a behavioral choice[43–45]. Further, lesioning or silencing of M2 produces alterations in this behavioral choice[46–49]. We therefore hypothesized that S2 connectivity with the M2 region might underlie the observed somatosensory hypersensitivity and allodynic state.

We first anatomically defined the neural identity of S2 to M2 projection neurons by injecting either a retrograde dye (CTB-555) or

sparse-labeling with retrograde AAV-tdTomato into M2 and performed RNAscope in situ hybridization and immunofluorescence using defined markers (Fig. 3h, top). The majority of S2-to-M2 neurons were located in layer V, specifically Va, with scattered neurons throughout layer II/III and layer VI (Fig. 3h, bottom). Using four markers of layer V neurons, we found that S2-to-M2 neurons are positive for the callosal intratelencephalic (IT) excitatory neuron markers *Trib2*, *Etv1*, and *Satb2*, but negative for *Ctip2*, a marker of layer Vb excitatory corticospinal neurons (Fig. 3n–q)[50–52]. Further, the scattered neurons within layer II/III were also found to be excitatory as they express *FoxP1*, a marker of cortical excitatory neurons (Fig. 3i–m)[53]. This agrees with tracing data from the Allen Brain Atlas in which anterograde tracing of excitatory layer V neurons in S2 contributes to the largest population

**Fig. 2 | The S2 cortical responses to mechanical and thermal somatosensory stimuli in the non-noxious and noxious range correlates with behavioral outputs. a** Diagram depicting the experimental strategy of in vivo calcium fiber photometry. GCaMP6f was injected into the S2 region of parvalbumin-cre (PV-Cre) mice, and a fiber lowered into S2 to capture calcium transients. **b** Example of GCaMP6f expression in PV neurons of the S2 region. Scale bar = 500 μm. **c** Percent withdrawal to differentially weighted mechanical von Frey stimulation of the hindpaw in fiber implanted animals both before and after inflammatory induction (*n* = 3 mice). **d–g** Left: Calcium responses of PV neurons in S2 to a 0.04, 0.16, 0.6 and 1.4 g mechanical stimulus following a single stimulation of the hindpaw at time 0. Blue shading represents the average temporal extent of stimulation. Right: Area under curve analysis of calcium transients produced in **d–g**. Two-tailed unpaired t-test. (*n* = 3 mice, 10 trials per mouse, for 0.6 g − *p* = 0.0066). **h** Time to peak of calcium response to a single mechanical stimulus of the hindpaw. Black circles and red circles represent no paw withdrawal and paw withdrawal trials, respectively.

**i** Left: Calcium responses of PV neurons in S2 to a heat ramp stimulus before and after zymosan induced inflammation. Time 0 is time of paw withdrawal from the heat source. Blue shading represents when the heat stimulus is on. Right: Area under curve analysis of the calcium transients produced during heat stimulus trials. (*n* = 3 mice, 5 trials per mouse) Two-tailed unpaired t-test. **j** Left: Calcium responses in PV neurons in S2 to application of acetone to the hindpaw at time 0. Blue shading represents the temporal extent of the acetone stimulus Right: Area under curve analysis of calcium transients produced during acetone application trials (*n* = 3 mice, 3 trials per mouse). **k–n** Left: Calcium responses of PV neurons in S2 following a single mechanical stimulation of the hindpaw at time 0, 4 h post zymosan injection. Force of stimulation (between 0.04–1.4 g) displayed in the graphs. Right: Area under curve analysis of calcium transients in **k–n**. Two-tailed unpaired t-test (*n* = 3 animals, 10 trials per mouse, for 0.16 g − *p* = 0.0014). Data as mean ± SEM. *$p$ = <0.05, **$p$ = <0.005, ***$p$ = <0.0005. See Source Data. Illustrations generated with Biorender.com.

of M2 projections with some projections from layer II/III[54] (Supplementary Fig. 5). S2 to M2 projecting neurons are, therefore, excitatory IT projection neurons, largely originating from layer Va.

## Connectivity reveals hierarchical relationship between S2 and M2

The layered structure of the cortex is organized in a manner to facilitate differential computations. Cortico-cortical communication pathways can stem from layer II/III to layer II/III or from deeper layer IT neurons targeting superficial layers[42,55,56]. Modulatory pathways typically innervate the superficial layers whereas driver pathways tend to innervate deeper layers[56,57] (but see ref. [58]). We set out to address which layer of M2 is innervated by S2 with monosynaptic rabies tracing using Cre drivers that express in either layer II/III (Penk-Cre) or layer V (Rbp4-Cre) (Fig. 4a). In this strategy, Cre-positive neurons act as "starter cells" that are selective hosts of rabies infection and retrograde monosynaptic transport. This identifies "input cells", thereby providing direct evidence of monosynaptic connectivity between two neurons (Fig. 4a). Viral injections into M2 successfully targeted layer II/III and layer V neurons, respectively, labeling many "starter cells", through GFP expression (Fig. 4b–d, g–i). Examination of the rabies virus mCherry + -labeled "input cells" in S2 compared to the total "starter cell" number in M2 revealed the majority of the S2-to-M2 projection neurons reside in deeper layers of S2 (notably layer V, in line with our previous retrograde tracing) and that they primarily innervate superficial layer II/III Penk+ neurons in M2 (Fig. 4e, f, j, k). This arrangement is in line with the role of these neurons playing a modulatory role on M2 activity. Recent M2 single cell sequencing studies have identified layer II/III Penk+ neurons as excitatory neurons[59,60]. This suggests that S2 provides modulatory excitatory input to M2 during somatosensation to govern stimulus-elicited behavioral choice.

## Inhibition of S2-to-M2 projecting neurons enhances sensitivity

To functionally manipulate S2-to-M2 IT neurons we used an intersecting viral strategy with injection of a retrograde AAV encoding Cre-recombinase into the M2 region along with an AAV encoding either a Cre-dependent excitatory (HM3Dq) or an inhibitory (hM4Di) chemogenetic receptor (designer receptor activated exclusively by designer drugs (DREADDs)) to either activate or inhibit S2-to-M2 projection neurons with the ligand clozapine n-oxide (CNO)[61,62] (Fig. 5a, b). CNO application rapidly and reversibly inhibited firing in S2-to-M2 projection neurons in slices from the inhibitory DREADD mice (Fig. 5c). Likewise, analysis of c-Fos expression as a marker of activity-dependent early immediate gene transcription demonstrated that CNO injection significantly upregulated c-Fos expression in animals injected with a Cre-dependent excitatory DREADD (HM3q) but not a mCherry virus (Supplementary Fig. 6a–d). This demonstrates our chemogenetic approach can increase or decrease S2-to-M2 neural activity efficiently.

Examining mechanical and thermal sensitivity in these animals revealed that injection of animals that express the inhibitory DREADD in the S2-to-M2 projection neurons with CNO phenocopied our behavioral observations with optogenetic inhibition of S2 by activation of PV+ inhibitory interneurons. Specifically, 30 min after CNO injection, S2-to-M2 inhibitory DREADD animals showed strong tactile hypersensitivity (hM4Di post-CNO contralateral (0.329 g ± 0.089) paw vs. ipsilateral (0.890 g ± 0.058) and mCherry contralateral (1.059 g ± 0.160) and heat sensitivity (hM4Di contralateral paw (5.505 s ± 0.4913) vs. ipsilateral (14.25 ± 2.119) and mCherry contralateral paw (16.53 s ± 2.627)) (Fig. 5d, e). Again, cold sensitivity remained unaffected (hM4Di contralateral 2.993 s ± 1.063 vs. mCherry contralateral 4.572 s ± 2.107) (Fig. 5f). Both the paw ipsilateral to the virus injection and animals infected with a control Cre-dependent mCherry virus showed no change in mechanical/thermal sensitivity, confirming that this effect is both limited to the targeted hemisphere and that viral infection is not a confound (Fig. 5d–f). Excitation of this circuit with an excitatory DREADD produced no effect on tactile sensitivity irrespective of the concentration of CNO used, suggesting that it is only the inhibition of this circuit that specifically governs sensitivity to somatic stimulation (Fig. 5d–f and Supplementary Fig. 5e).

To confirm that inhibition of the input from axons projecting from S2 into the M2 cortical region is responsible for the increase in mechanical and heat sensitivity, we inserted cannulas into M2 in inhibitory DREADD animals, and locally microinjected CNO (as used in refs. [63,64]) (Fig. 5g, h). Local microinjection of CNO (300 nl of 300μM) in M2 increased behavioral mechanical sensitivity similar to that produced by systemic (i.p. 3 mg/kg) injections (300 μM CNO 0.4778 g ± 0.104 vs. Saline 0.9243 g ± 0.115) (Fig. 5i) and also increased heat sensitivity (300 μM CNO 6.031 s ± 0.641 vs. Saline 10.48 s ± 0.703) (Fig. 5j). Importantly, local injection of CNO into control animals without viral expression had no effect on mechanical or heat sensitivity (Supplementary Fig. 7a, b). This confirms that it is the output from S2 to M2 that influences mechanical and heat nociceptive threshold sensitivity.

The secondary motor cortex is suggested to be involved with the planning of motor behavior in addition to modulating behavioral responses[48,65]. The increased withdrawal response sensitivity to mechanical and thermal stimulation observed on inhibiting S2 projections to M2 could therefore be due to an altered gross motor reactivity, as shown for S2-to-S1 projecting neurons[66]. To ascertain whether motor behavior was significantly altered by changing activity of S2-to-M2 neurons, we analyzed gait and motor function of mice infected with DREADDs. Examining the sciatic functional index (SFI) to compare the position of one hindpaw (DREADD affected) to the other (control) during locomotion, we found no significant differences in their locomotory behavior (Supplementary Fig. 7c). There was also no change in stride length (Supplementary Fig. 7d). This suggests that the S2 to M2 circuit is primarily sensory in nature and exerts its effects on

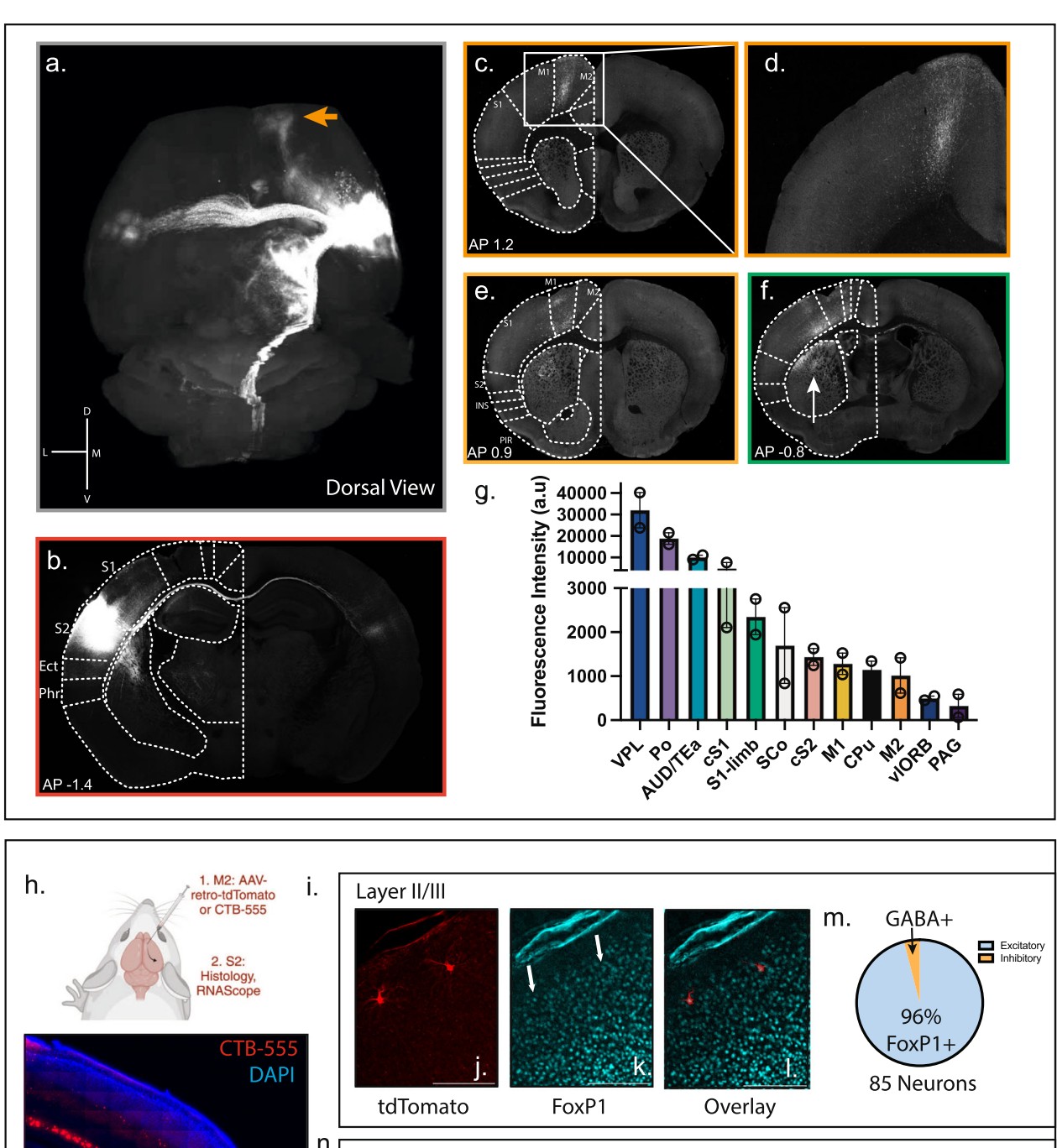

an animal's response to sensory stimulation independent of gross motor alteration, complex decision making, or learning.

## Discussion

The central neural substrates of somatosensory behavior and how they work together to orchestrate both simple and complex sensory experiences, is still largely unknown. Top-down circuits, often higher order central to lower order central/peripheral circuits, are important modulators of behavior. In somatosensory circuits, top-down modulation can occur from cortical circuits (primary somatosensory cortex (S1), anterior cingulate cortex (ACC)) and subcortical circuits (amygdala and brainstem)[4,67–69]. We previously characterized a S1 excitatory

**Fig. 3 | The S2 projectome reveals specific cortical and subcortical sensorimotor targets including the secondary motor cortex (M2). a** Dorsal view of the reconstruction of S2 projection sites. Tract from S2 to M2 highlighted by an orange arrow. **b** Example S2 injection site. Scale bar: 1 mm. **c** M2 projections Scale bar: 1 mm. **d** magnified M2 projections. Scale bar: 500 μm. **e** Projections to the M1 cortex. Scale bar: 1 mm. **f** Projections to the primary somatosensory cortex (S1). Arrow denotes projections to the striatum. Scale bar: 1 mm. **g** Average intensity of projections to different brain regions. S2 – contralateral light red, S1 – contralateral light green, vlORB – navy, M1 – yellow, M2 – orange, AUD/TEa – teal, caudate putamen – black, Po of thalamus – purple, VPL of thalamus – purple, superior colliculus – white, PAG – dark purple. *n* = 2 animals, averages from 3 slides taken. **h** Top: Schematic for ascertaining the identity of S2-to-M2 neurons by injecting either the retrograde dye CTB-555 or AAV2/retro-CAG-tdTomato into M2 and

analyzing the S2 region generated with Biorender.com. Bottom: Representative image of the S2 region labeled with CTB-555. Scale bar: 500 μm. **i** Identity of Layer II/III neurons. **j** Image of retro-CAG-tdTomato sparse labeled neurons in layer II of S2. **k** Foxp1 staining in the cortex as a marker of excitatory neurons. Arrows pointing to example positive Foxp1 neurons. **l** Merged image reveals overlap between retro-CAG-tdTomato labeled neurons and Foxp1. Scale bar: 100 μm. **m** Quantification of excitatory vs. inhibitory Layer II/III neurons. Analysis of 3 animals with 85 neurons total. **n** Identity of Layer V neurons by RNAscope. **o** Example of non-overlap between CTB-555 (red) labeled neurons and Ctip2 RNAscope probe (green). **p** Example of overlap between CTB-555 (red) labeled neurons and Trib2 (green). **q** Quantification of overlap between CTB-555 labeled neurons and Ctip2, Etv1, Satb2, and Trib2. *n* = 2 animals. Data presented as mean ± SEM. Scale bar: 50 μm. See Source Data.

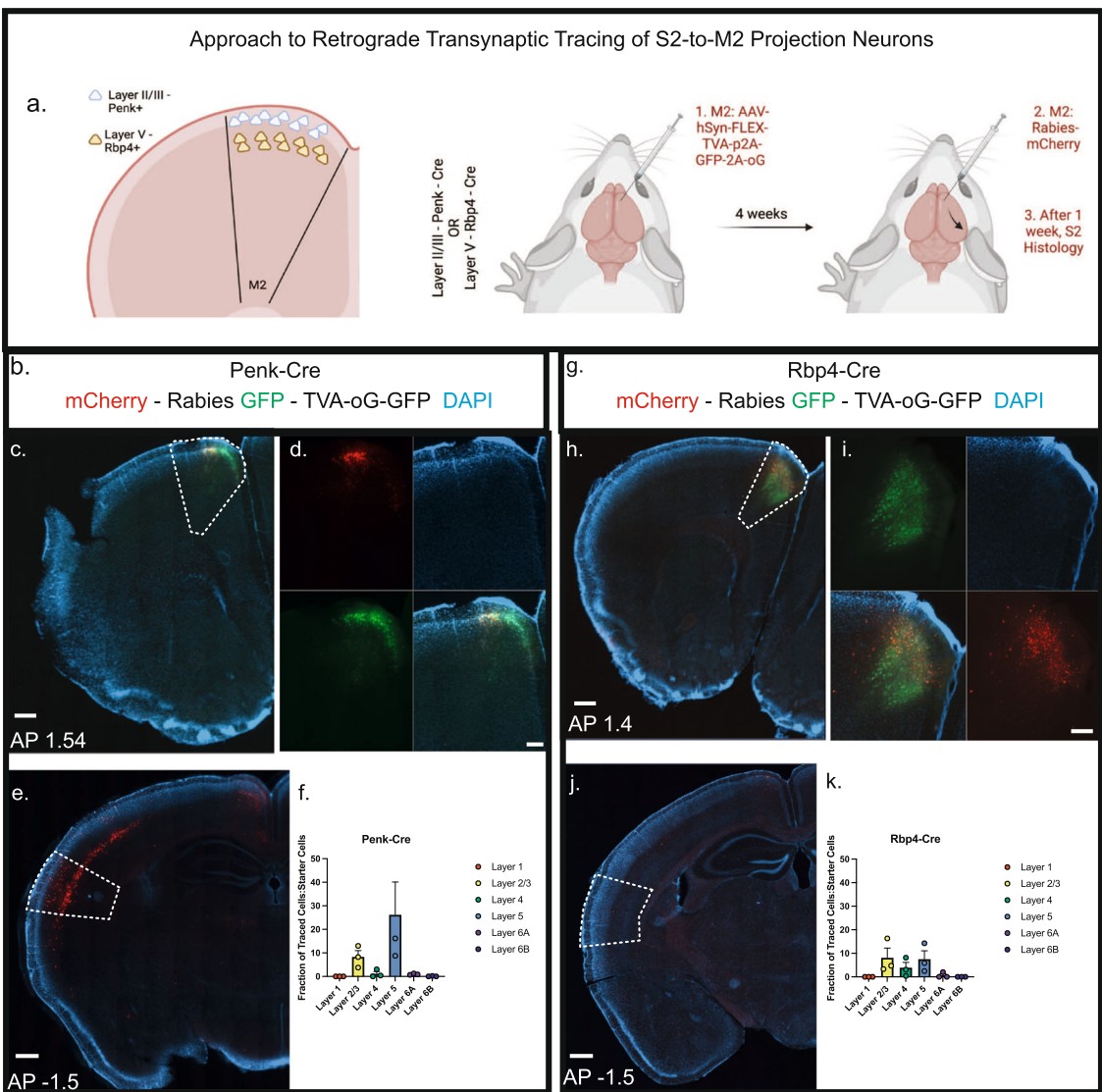

**Fig. 4 | S2 monosynaptically connects with upper layer II/III neurons in M2.**
**a** Schematic diagram of layer specific rabies tracing. **b** Example of M2 injection site in Penk-Cre animal. GFP: AAV-hSyn-Flex-TVA-p2A-GFP-2A-oG. mCherry: Rabies. Blue: DAPI. **c** Widefield view of M2 with M2 region outlined. Scale bar = 500 μm. **d** Zoomed in view divided by channel. Scale bar = 200 μm. **e** Widefield view of S2 with S2 region outlined. **f** Layer-specific quantification of the percentage of "starter cells (GFP+)" in M2 compared with "input neurons (mCherry+)" in S2 in Penk-Cre

Mice. **g** Example of M2 injection site in Rbp4-Cre animal. GFP: AAV-hSyn-Flex-TVA-p2A-GFP-2A-oG. mCherry: Rabies. Blue: DAPI. **h** Widefield view of M2 with M2 region outlined. Scale bar = 500 μm. **i** Zoomed in view divided by channel. Scale bar = 200 μm. **j** Widefield view of S2 with S2 region outlined. **k** Layer-specific quantification of the percentage of starter cells in M2 compared with traced neurons in S2 in Rbp4-Cre Mice. Data presented as mean ± SEM. For all experiments, *n* = 3 mice. See Source Data. Illustrations generated with Biorender.com.

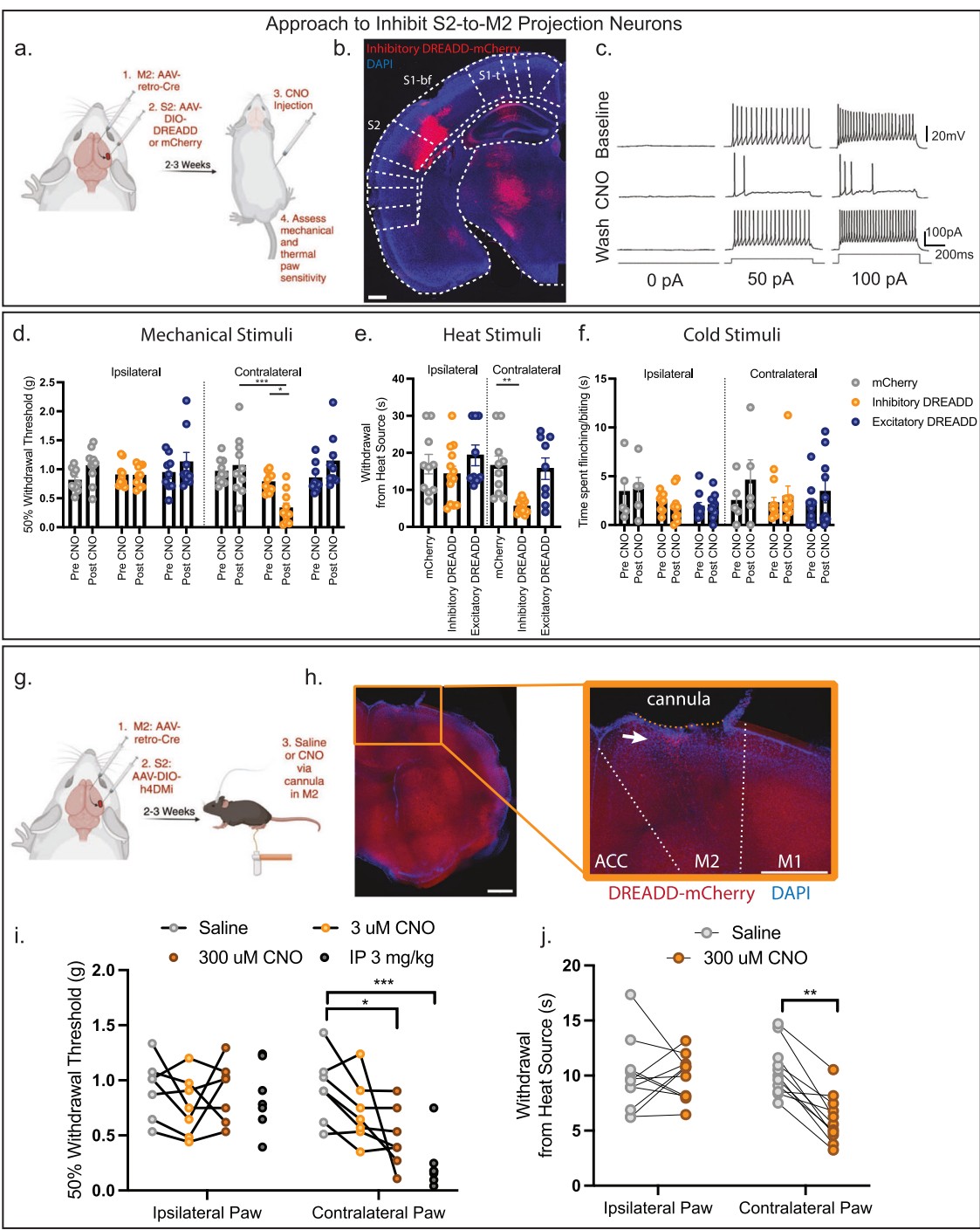

**Fig. 5 | Chemogenetic inhibition of S2-to-M2 projecting neurons enhances mechanical and heat sensitivity. a** Schematic diagram and timeline of chemogenetic manipulation of S2-to-M2 projecting neurons. **b** Representative image of S2 region following injection AAV-DIO-Inhibitory DREADD in S2 and AAV2/retro-Cre in M2. Expression can be observed in layer V and VI, as well as scattered expression in layer II/III. Scale bar = 500 µm. **c** CNO ligand administration in ex vivo brain slices during patch clamp recording inhibits neurons infected with inhibitory DREADDs. Example trace representative of 4 animals (1 neuron/animal) examined. **d** Chemogenetic inhibition of S2-to-M2 neurons produces tactile sensitivity in the von Frey assay. mCherry: $n = 10$, Inhibitory DREADD: $n = 11$, Excitatory DREADD: $n = 10$. Two-Way ANOVA followed by Tukey's multiple comparisons test mCherry contralateral vs. Inhibitory DREADD contralateral $p = 0.0002$. **e** Chemogenetic inhibition of S2-to-M2 neurons produces heat sensitivity in the Hargrave's assay mCherry: $n = 10$, Inhibitory DREADD: $n = 11$, Excitatory DREADD: $n = 10$. Kruskal–Wallis H = 25.08 $p = 0.0001$ Dunn's multiple comparisons mCherry contralateral vs. Inhibitory DREADD contralateral $p = 0.0032$. **f** Chemogenetic

inhibition or excitation of S2-to-M2 neurons produces no effect on acetone-induced cold sensitivity mCherry: $n = 5$, Inhibitory DREADD: $n = 6$, Excitatory DREADD: $n = 10$ Two-Way ANOVA followed by Tukey's post-hoc test. **g** Schematic diagram of chemogenetic inhibition of M2 projections via cannula administration of CNO. **h** Representative image of cannula placement above M2. White arrow pointing towards S2-to-M2 axons in the cortical column. Scale bar = 500 µm. **i** Chemogenetic inhibition of local M2 projections reproduces the tactile sensitivity observed with systemic CNO application. mCherry: $n = 8$, inhibitory DREADD: $n = 8$ Contralateral Paw Saline vs. 300 µM CNO Two Way ANOVA followed by Tukey's. Saline vs. 300 µM CNO - $p = 0.0222$. Saline vs. 3 mg/kg CNO i.p. – $p = 0.0002$. **j** Chemogenetic inhibition of local M2 projections reproduces the heat sensitivity observed with systemic CNO application. mCherry: $n = 8$, inhibitory DREADD: $n = 8$. Contralateral Paw Saline vs. 300 µM CNO Two Way ANOVA followed by Sidak's. Saline vs. 300 µM CNO $p = 0.0003$. Data presented as mean ± SEM. *$p = < 0.05$, **$p = < 0.005$, ***$p = < 0.0005$. See Source Data. Illustrations generated with Biorender.com.

corticospinal circuit in which inhibition produces decreased mechanical sensitivity[4]. However, this circuit failed to respond to heat or alter heat-evoked thermosensory behaviors[4]. Indeed, recent work has confirmed this, showing that S1 primarily encodes cooling but not heating[6]. We now show that S2 can fulfill the role of heat encoding and modulate mechanical responses in a distinct manner from S1, through a corticocortical circuit.

Exactly which modalities and magnitudes of somatosensory information S2 responds to and modulates remains an open question. Data in rodents[70–72] and primates[73–75] have identified S2 as responsive to tactile, heat, and cold temperatures but whether it responds to both noxious and non-noxious information remains unclear, especially in human functional imaging studies[19,26]. To address this, we performed fiber photometry recordings in both PV inhibitory neurons and CaMKII excitatory neurons during somatosensory stimulation. Fiber photometry recordings demonstrate that S2 PV neurons can respond to mechanical, heat, and cooling stimuli applied to the hindpaw. Interestingly, in contrast, extensive variability exists in the recordings of CaMKII excitatory neurons to different mechanical stimuli, with an optimal response obtained at 0.6 g stimulation. With such high variability, we hypothesize that the organization of mechanical information in S2 may be scattered with respect to force encoding, similar to the organization of visual information in the primary visual cortex and vibrotactile stimuli in S1[32,33]. In contrast, a much more coherent response was obtained with a ramping heating stimulus. Comparing the dynamics of the two revealed that overall, PV inhibitory interneuron activity rises first followed by excitatory neurons, in line with PV neurons being fast-spiking. The exact excitatory/inhibitory balance between these two populations will require further studies utilizing single-cell imaging or electrophysiological approaches to resolve exactly how these dynamics alter behavioral reactivity and how S2 is organized with respect to modality.

Interestingly, optogenetic inhibition of S2 produces mechanical and heat behavioral hypersensitivity without affecting cooling sensitivity. These conclusions are supported by chemogenetic inhibition of specific S2 neurons that replicates this effect. This suggests a difference in S2 in encoding vs. behavioral modulation. Alternatively, our findings on cooling sensitivity stem from use of the acetone assay in which a drop of acetone is applied to the paw to induce evaporative cooling and measuring the amount of time the mouse flinches, flicks, licks, or bites their paw. This is different than examining the threshold of response as in our mechanical and heat assays. It may be that the response threshold to cooling is altered, however, there is no technical way to apply a consistent ramping cold stimulus to a single paw in freely-moving mice.

Other work has identified the posterior insular cortex as a crucial substrate of thermosensory behaviors, responding to both cooling and heating[6]. Accordingly, optogenetic inhibition of this region produces deficits in detecting thermal changes in a Go/No-Go task[6]. While close in anatomical space and sharing some connectivity, whether they interact to achieve a common goal or act on common downstream elements to exert an influence on behavior is an interesting future direction. It may be that S2 gates the thermal threshold for behavioral response while the insular cortex is able to extract information on the magnitude of temperature change. Alternatively, as the study on the posterior insula focused on studying non-painful temperatures and our study examined behaviorally noxious stimuli, this may suggest a difference in encoding of low and high threshold thermal stimuli at the cortical level, suggested in some human studies[26]. In line with this, electrophysiological recordings in the rat posterior triangular thalamic neurons that project to S2 respond to noxious but not non-noxious mechanical and heat stimuli, similar to our responsivity profile in S2[76]. The emerging neural architecture highlighted by this study and others[4,6], favors a model in which different somatosensory modalities are processed in distinct cortical regions.

Previously defined somatosensory cortical circuits which mitigate sensorimotor action originate from the primary somatosensory cortex (S1)[4,77] and exert action via direct or indirect spinal connections or through connections with the primary motor cortex (M1)[78], or from S2 to S1/M1[13,14,40]. Indeed, silencing of S2 to S1 connections produces locomotor deficits such as hindpaw angle[66]. However, we now find that the secondary motor cortex (M2) is an important cortical substrate that S2 acts through and that this is distinct of the S2 to S1 circuit previously described as its manipulation does not impact hindpaw locomotor behavior. Chemogenetic inhibition of S2-to-M2 neurons phenocopies the sensory effects of broad S2 inhibition and local inhibition of S2 axons that project into M2 also increase mechanical and heat sensitivity. M2 (also known as the supplementary motor area (SMA) or agranular cortex (AGm)) is part of the rodent prefrontal cortex and involved with motor planning, choice-based behavior, and motor learning[48,65,79,80]. Indeed, lesioning of M2 in rodents produces reduced performance in many sensory-based Go/No-Go tasks[45–47,81]. Further, recording neural activity in rats trained in a modified T-maze in which each arm has different probabilities of reward has revealed that M2 displays the earliest choice-related activity of any cortical region[46]. Similarly, work in the primate and the mouse whisker system has identified S2 as important for sensory-based decision making[14,74]. Specifically, S2 to S1 projections are thought to be important in the encoding of choice following a stimulus[14]. It is likely that these neural substrates (S1, S2, and M2, in conjunction with the insula and M1) work together to process sensory information and produce accurate behavioral responses. Our anatomical tracing data favors a model where M2 is a higher-order structure that would receive sensory information from multiple sources before exerting an effect on behavior. Indeed, in multisensory Go/No-Go behavioral tasks, M2 optogenetic inactivation produced behavioral deficits in responses to an audiovisual cue and recording in M2 neurons suggests that it accumulates sensory information from multiple sources to make decisions[82]. However, our behavioral assays are intrinsically different in that the choice (paw withdrawal or not) is not a learned behavior nor coupled to any predictable stimulus and is reflexive in nature. This indicates that S2 inputs to M2 may gate an animal's behavioral response to defined somatosensory stimuli, in addition to its more complex roles in sensory decision making, choice, association, and learning. Our anatomical tracing supports this theory by demonstrating that layer Va pyramidal neurons in S2 predominantly provide input to layer II/III neurons in M2, a connectivity pattern typically associated with modulating rather than driving cortical responses[56].

Taken together, this places S2/M2 circuitry as a core mediator of somatosensory behavior. How this circuit works in collaboration with other defined cortical circuits is an interesting future direction, but we hypothesize since distinct areas of the somatosensory cortex appear to process distinct modalities that these circuits operate in a parallel fashion to provide a comprehensive picture of the somatosensory environment.

## Methods

### Animals

Both male and female C57BL6/J (Jax #000664) and B6.129P2-Pvalb[tm1(cre)Arbr]/J (Jax #017320) mice of roughly equal numbers (see Source Data) were used for the behavioral and imaging experiments unless otherwise stated, see figure legends for details. Mice were injected with AAVs between 10–14 weeks of age and behavior was conducted between 3–10 weeks post injection, depending on viral expression. All mice in this study were kept on a 12 h light cycle, at 21–23 °C, with 30–50% humidity.

### Inclusion and ethics

All experiments were conducted in a blinded fashion with strict accordance to the guidelines set forth by the Boston Children's

Hospital Institutional Animal Use and Care Committee under protocols 00001507, 00001546, and 20-05-4165.

## Stereotaxic injection of adeno-associated viruses and fiber/cannula implantation

3% of isoflurane was used for induction and 1–3% isoflurane for maintenance of anesthesia. Hair was shaved and the surgical sterilized with betadine and ethanol before an incision to expose the skull was made. All coordinates were identified relative to the bregma on the skull. Coordinates for the hindlimb secondary somatosensory cortex (S2) (ML: 3.9 AP: −1.3 DV: 2.5) were determined based upon our previous anatomical studies[4]. Secondary motor cortex (M2) coordinates (ML: 0.4 AP: +1.34 DV: 0.8) were chosen based upon the densest projection site from S2 in preliminary anatomical tracing studies. Primary somatosensory barrel field coordinates (ML: 2.9 AP: −1.3 DV: 1.5) were located 1 mm medially from S2 injections to rule out the effect of viral spillover into S1. All injection sites were verified post-mortem.

For optogenetic stimulation, surgery proceeded as above with viral injection but in addition 2 mm, 200 μM diameter 0.39NA optical fiber cannula (Thor Labs CFMLC12L05) and for fiber photometry, 3 mm cleaved, 200 μM diameter 0.37NA black ceramic fiber optic cannulae (Neurophotometrics Ltd.) were implanted into the S2 region and affixed to the skull with dental cement. Two skull screws (Fine Science Tools 19010-10) were implanted on the opposite hemisphere of the fiber cannulae, roughly above the coordinates of the primary motor cortex and primary visual cortex.

For cannula studies, surgery proceeded as above with viral injection but in addition a 0.8 mm cannula was implanted in the M2 region and affixed to the skull with dental cement. Skull screws were used to stabilize the implant as above. A cap that consisted of a 1 mm dummy cannula was used when the injector unit was not in use.

For rabies tracing studies, animals were anesthetized with ketamine(120 mg/kg)/xylazine(10 mg/kg) and the skull exposed as above. Either AAV2/9-Syn-Flex-TVA-oG-GFP or AAV2/8-Syn-Flex-TVA-oG-GFP was injected into the secondary motor cortex of either Penk-Cre (Jax # 025112) or Rbp4-Cre (MGI:4367067) animals at a depth of 200 (to target layer II/III) or 400 micrometers (to target layer V), respectively. Six weeks post-injection, SADdg-EnvA-mCherry was injected at four points along M2 to capture a wide breadth of starter cells. Animals were taken for histology one week following.

Viruses used in this study include: AAV2/9.Syn.-Flex.GCaMP6f.WPRE.SV40 (1E + 13 Addgene 100833 – 100 nl), AAV2/1-CAG-FLEX-rev-ChR2-tdtomato (1.23E + 13 gc/mL – Boston Children's Hospital Viral Core – 125 nl), AAV2/retro-CAG-Cre-WPRE (2.89E + 13 gc/mL - Boston Children's Hospital Viral Core – 125 nl), AAV2/1-Syn-DIO-hM4Di-mCherry (Boston Children's Hospital Viral Core – 125 nl), AAV2/9-hSyn-DIO-Hm3D(Gq)-mCherry (6.21432E + 13 gc/mL - Boston Children's Hospital Viral Core – 125 nl), AAV2/9-hSyn-DIO-mCherry (Boston Children's Hospital Viral Core) – 125 nl, AAV2/9-CAG-tdTomato-WPRE (1.0234E + 13 gc/mL Boston Children's Hospital Viral Core – 50 nl), AAV2/retro-hSyn-tdTomato-WPRE (Boston Children's Hospital Viral Core – 125 nl), SADdg-EnvA-mCherry (2.26E + 08 to 4.10E + 10 TU/mL Boston Children's Hospital Viral Core), AAV2/9-Syn-Flex-TVA-oG-GFP (1e13 gc/ml Boston Children's Hospital Viral Core) AAV2/8-Syn-Flex-TVA-oG-GFP (1e13 gc/ml Boston Children's Hospital Viral Core), AAV2/9-CamKII-GCaMP6f-WPRE-SV40 (Addgene 100834-AAV9 – 125 nl).

## Electrophysiology

To obtain brain slices containing S2, mice were anesthetized using isoflurane and decapitated into oxygenated (95% O2; 5% CO2) ice-cold cutting solution (in mM): 130 K-gluconate, 15 KCl, 0.05 EGTA, 20 HEPES, and 25 glucose (pH 7.4 adjusted with NaOH, 310–315 mOsm). The brain was then removed quickly and immersed in the ice-cold cutting solution for 60 s. Coronal slices containing S2 were sectioned and collected. The brain was cut with a steel razor blade, then sectioned into 250-μm-thick slices in the oxygenated ice-cold cutting solution using a sapphire blade (Delaware Diamond Knives, Wilmington, DE) on a vibratome (VT1200S; Leica, Deerfield, IL). The slices collected were allowed to recover at 30 °C for 20 min in oxygenated saline solution (in mM): 125 NaCl, 26 NaHCO3, 1.25 NaH2PO4, 2.5 KCl, 1.0 MgCl2, 2.0 CaCl2, and 25 glucose (pH 7.4, 310–315 mOsm).

To detect the inhibitory innervation of PV interneurons on pyramidal neurons in S2 L2/3 or L5, ChR2-mCherry was expressed in PV interneurons and pyramidal neurons were recorded in response to stimulation of PV interneurons. Pyramidal neurons in S2 L2/3 and L5 were visualized through a monitor with projection from the camera of a DIC-equipped microscope (Prime BSI, Teledyne Photometrics). Inhibitory post-synaptic currents (IPSCs) were obtained by holding the membrane potential at 0 mV and with the presence of NBQX (10 μM, 0373, Tocris) and CPP (20 μM, 0247, Tocris) to block AMPA and NMDA receptors, respectively. Evoked IPSCs (eIPSCs) were induced by applying a train of single pulses (0.2 ms) of full-field illumination of blue light through the 60× objective (Olympus LUMplanFL N 60×/1.00 W) with interval of 250 ms. The blue light (470 nm, 83 mW/mm²) was supplied by a CoolLED pE unit. Spiking of pyramidal neurons were detected through current clamp mode by injecting currents ranging from 100 to 600 pA. The effect of PV inhibitory transmission on the firing of pyramidal neurons was examined by stimulating PV interneurons using blue light at a frequency of 40 Hz with each pulse lasting for 20 ms. Electrophysiological data acquisition and offline analysis were performed using custom software in IgorPro (Wave-Metrics, Portland, OR). eIPSCs were averaged from 3 to 5 traces.

To detect the expression of inhibitory DREADD, pyramidal neurons in S2 L2/3 or L5 projecting to M2 were recorded through whole-cell patch clamp and tested with CNO. The pyramidal neurons were labeled by mCherry and can be visualized through fluorescence microscope (Olympus, Japan). Glass pipettes (Drummond Scientific) were pulled on Sutter p87 Flaming/Brown micropipette puller (Sutter Instruments) and filled with internal solution containing (in mM): 150 K-gluconate, 8 KCl, 10 EGTA, 10 HEPES (pH7.3, 290–300 mOsm) to optimize the pipette resistance to be 3.5-4.0 MOhm. Patch recordings were performed using a MultiClamp 700B (Axon Instruments, Foster City, CA) and digitized at 20 kHz with an ITC-18 interface (Instrutech). Intrinsic cellular properties of pyramidal neurons were measured in current clamp mode. I-V curves were obtained by recording firing rates when using current injection from 0 to 600 pA in steps of 50 pA with duration of 1 s. CNO (5 μM, BML-NS105, Enzo) was then perfused into the bath solution. More I-V curves were collected during (20 min since CNO perfusion) and after the washing out of CNO. Four repeats were conducted from four mice.

## Histology

Mice were anesthetized with 200 mg/kg pentobarbital and perfused transcardially with 4% paraformaldehyde (PFA) in PBS. Brains were isolated and fixed overnight in 4% PFA before storage in 1 X PBS. Brains were sectioned with a vibratome between 60–100 μm or a cryostat at 30 μm and mounted on slides (Fisher Permafrost). Vibratome sections were permeabilized with 1 X PBS with 0.2% Triton-X100, mounted on slides, and coverslipped with mounting media containing DAPI. All injection sites were aligned back to the Allen Brain Atlas, injection sites with substantial off-target infection were excluded. In the cannula experiments, signal from the AAV2/9-CAG-DIO-DREADD(h4Dmi) was amplified using anti-mCherry (Abcam: ab167453) at 1:500 by first blocking with 1 X PBS with 10% goat serum and 0.3% Triton X-100 for one hour at room temperature, incubated with anti-mCherry overnight at 4 °C, followed by incubation with goat anti-rabbit 568 (Thermo Scientific: A-110011) for one hour at room temperature, and cover slipped with DAPI mounting media. For identification of excitatory cortical neurons in layer II/III, anti-Foxp1 (Abcam: ab16645 1:500) was used with goat anti-rabbit 488 (Thermo Scientific: A-11008). Slides were visualized and imaged with a Nikon Ti-1200.

## RNAscope

Injections of cholera toxin subunit b conjugated to Alexa Fluor 555 (CTB-555) (Thermo Scientific C34776) was injected at 1 ×150–200 nl in the secondary motor cortex as described above. Two weeks following injection, animals were processed as above but after incubation with PFA were placed into 30% sucrose in 1 X PBS for 2 days. Brains were rinsed in 1 X PBS and snap frozen in optimal cutting temperature compound (TissueTek 4583) and stored at −80 °C until processing. Brains were sectioned with a cryostat at 20 micrometers. RNAscope was performed per manufacturers instructions (ACDBio – RNAscope Multiplex Fluorescent V2 Assay). Probes against Ctip2 (413271-C1), Etv1 (557891-C2), Trib2 (514021-C1), and Satb2 (413261-C2) were used and slides coverslipped with DAPI mounting media. Slides were imaged with Olympus VS120 SlideScanning microscope.

## Fiber photometry

Mice were habituated to the fiber optic cable for one hour each on two separate days. Photometry signals were acquired by alternative illumination with 470 nm (GCaMP) and 410 nm light (isobestic control) (Neurophotometrics FP3001). Any mice with calcium responses that occurred during whisker deflection (suggesting fiber was placed in barrel cortex) or sound (suggesting fiber was place in adjacent auditory cortex) were excluded and only animals with responses to hindpaw stimulation were used. Tactile stimuli, heat stimuli, and cold stimuli were presented 10, 5, and 10 times per mouse separated by at least 10–30 s in the case of tactile and 1 min in the case of thermal. In short, the raw delta F/F is first smoothened and the slope due to fluorescent decay removed with the airPLS algorithm. The signals are then standardized and Z-scored. For von Frey and acetone, averaged traces across animals were aligned to the point when the stimulus was applied. For heat stimuli via Hargreaves' (see below), traces were aligned to the paw withdrawal event, as a ramping heat stimulus until a paw withdrawal in a freely-moving mouse can produce variable times of heat exposure (see Fig. 1g for example of variation). Acquisition was orchestrated by Bonsai software (https://bonsai-rx.org/) in which a behavioral camera (Microsoft LifeCam HD-3000) and the fiber photometry acquisition were triggered simultaneously. Fiber photometry data was aligned back to the behavioral video recordings by assigning each frame a timestamp. For area under curve (AUC) analysis, the first 10 frames of von Frey (between 0.42–3 s) and 31 frames of Hargreaves' and acetone were averaged and used as the baseline. Both positive and negative areas were determined and subtracted to produce a net area reflective of the total summarized amplitude throughout the trial. Only peaks greater than 10% of baseline were analyzed.

## Von frey mechanical sensitivity assay

Mice were placed in individual square chambers on grated flooring to allow access to the hindpaw from the bottom. Mechanical thresholds were acquired using the Up-Down method in which filaments that exert known forces are applied to the hindpaw in a successive order to ascertain the 50% Withdrawal Threshold. Mice were allowed to habituate to the chamber for 1 h on two separate days in which 5 stimulations of each of the hindpaws with a 0.6 g filament was applied to get the mice accustomed to foot stimulation. On the day of sensory testing, mice were allowed to habituate in the chambers for 1 h before data acquisition. Hindpaw withdrawal was quantified as a fast, upward withdrawal from the stimulus, rapid paw shaking, biting, or licking, independent of any ongoing locomotion.

In a subset of animals (PV ChR2 and mCherry controls), the percentage of withdrawals to a range of mechanical stimuli (0.04–1.4 g) was assessed in order to differentiate between allodynic and hypersensitivity.

## Hargreaves' thermal assay

Heat sensitivity was assessed by placing mice in individual square chambers on a glass surface heated to 30 °C. A radiant heat source targeted to the hindpaw was applied until a withdrawal response or a 30 s cut off was reached. Withdrawal was quantified as a rapid removal of the paw from the heat source, shaking, licking, or biting of the paw was also considered an aversive response. Mice were allowed to habituate to the chamber for 1 h on two separate days and 1 h before on the day of testing.

## Acetone cold sensitivity assay

Cold sensitivity was determined by placing individuals in square chambers on grated flooring in order to access the hindpaw from below. Mice were allowed to habituate to the chamber for 1 h on two separate days. A 30 µL drop of acetone was placed on the surface of either the ipsilateral or contralateral paw and the amount of time the mouse spent biting or flinching the paw was quantified.

## Optogenetic activation of inhibitory parvalbumin neurons

Mice were habituated on two separate days for one hour each with a rotary fiber optic cable connected (Thor Labs RJPFL2) via a ceramic adapter to the optic fiber implant. Blue laser light pulses (470 nm, 40 Hz, 3 mW output) were administered during the duration of the experiment to activate PV neurons with a maximum exposure of 1 min per test.

## Thermal conditioned place preference assay

Two thermal plates (Bioseb BIO-CHP) forming two separate arenas (165 mm × 165 mm) were enclosed in an acrylic box with one side colored black and the other black-and-white striped. A black door allowed separation between the two arenas during training. Mouse tracking was enabled by an overhead camera that tracked the center point of the mouse and % of time spent in each arena was tracked and calculated (Ethovision). Each plate was set to 39 °C. Three weeks post-surgery, mCherry controls and ChR2 injected mice were assigned randomly to receive blue light stimulation (3 mW, 40 Hz, 1 min on/1 min off for 30 min) in either the black or white striped chamber. Day 1, mice were habituated to the entire apparatus for one hour. Day 2, mice were pre-tested to determine baseline preference for 15 min. Any mouse with a pre-preference of >80% was excluded. Day 3–6, mice were trained to associate the stimulus with the chamber by restricting to one chamber and received stimulation in the paired chamber or did not in the unpaired chamber for 30 min. On Day 7, place preference was determined by the mouse's % of time spent in either chamber in a 15-min time window.

## Elevated plus maze

An elevated maze consisting of two open arms, two closed arms, and a center arena was used to assess anxiolytic behaviors as in[83]. Briefly, mice were placed in the center arena and recorded using Ethovision XT software (Noldus) for 9 min. The first three minutes were the baseline period with no light, 3–6 min consisted of blue light stimulation at the same parameters described above, and 6–9 min the light stimulation was stopped to assess any after effects. Percentage of time spent in each arm was calculated and statistically compared.

## Analysis of motor behaviors

Motor behaviors in chemogenetic manipulated mice were recorded via Digigait. In brief, mice were placed on an illuminated treadmill with a camera below to capture paw placement. The treadmill was set to 20 cm and mice allowed to walk for a 5 min period. Sciatic functional index and stride length were calculated with the Digitgait Noldus software.

## Serial mapping of S2 projection targets

Two mice with 50 nl injections of AAV2/8-CAG-tdTomato in S2 were used for serial two photon tomography mapping. Mice were anesthetized with 200 mg/kg pentobarbital and perfused transcardially with 4% paraformaldehyde (PFA), brains isolated and incubated in 4%

PFA overnight at 4 °C. Brains were then embedded in 4.5% oxidized agarose and coronal sections of 1.38 um^2 resolution at 50 um optical section were taken using the TissueCyte (Tissue Vision Inc.). Images were aligned back to the Allen Brain Atlas using Neuroinfo software (MBF Bioscience) and projections were manually annotated and quantified using ImageJ. One mouse with a 100 nl injection of AAV2/8-CAG-tdTomato in S2 was processed traditionally with a vibratome and imaged with an IXM Confocal microscope to confirm the two photon findings.

### Clozapine-N-oxide administration during behavioral tests

To activate DREADD chemogenetic receptors in behavioral tests, 3 mg/kg clozapine-n-oxide (CNO – Enzo Biosciences BML-NS105) (first dissolved in DMSO and saline added until 0.02% DMSO final solution) was injected intraperitoneally 30 min prior to behavioral measurement. All behavioral measurements were separated by at least 24 h to ensure metabolism of CNO.

### Clozapine-N-oxide local administration during behavioral tests

For local administration of CNO in M2 via cannula, a 33-gauge injector cannula was attached to the cannula pedestal (P1 Technologies C315GS). Mice were habituated to the injector cannula for 2 days prior to behavioral assessment for 30 min each. On the day of testing, mice were habituated to the cannula injector for 5 min, 300 nanoliters of saline or different concentrations of CNO injected at 100 nl/min, and diffusion allowed to proceed for 2 min before behavioral assessment.

### Validation of PV-ChR2 and excitatory DREADD effects via cFos staining

Six mice with S2-to-M2 neurons labeled with either mCherry or Excitatory DREADD (three per condition) were injected with 3 mg/kg CNO. Animals were placed in the dark for one hour before perfusion. Histology proceeded as detailed above.

For validation that our light stimulation paradigm can inhibit pyramidal neurons in PV-ChR2 animals, six mice were injected with ChR2 or mCherry virus (three each) and fibers implanted. Stimulation occurred as in the place preference assay (3 mW, 40 Hz, 1 min on/1 min off for 30 min). Animals were placed in the dark one hour before perfusion. Staining proceeded as detailed above. The S2 region was then imaged with a Leica SP8 Confocal Microscope. Cells positive for c-fos within layer V of the area of viral injection (as established by expression of tdTomato) were counted across 2-3 serial sections, averaged, and compared statistically.

### Zymosan injection to induce peripheral inflammation

Zymosan (20 μl 5 mg/mL Sigma: Z4250) was injected into the hindpaw and GCaMP fiber photometry was performed as above during von Frey and Hargreave's assays at baseline and the peak of sensitivity (4 h post injection).

### Statistics and reproducibility

Statistics were performed in GraphPad Prism version 9. Exact tests performed are listed in the figure legends. For the entire manuscript, significance is defined as *$p$ = <0.05, **$p$ = <0.005, ***$p$ = <0.0005 unless otherwise stated. The PV-mCherry/ChR2 behavioral experiments presented in Fig. 1g, i, j, n were repeated independently 3 times with 3-4 animals per group with similar results. The fiber photometry experiments in Fig. 2 were repeated independently 3 times with 1 animal per group with similar results. The anatomical tracing experiments in Fig. 3f, were performed twice independently with two-photon tomography and once with serial histology with similar results. Retrograde tracing in Fig. 3g/h and g/i, was performed 4 independent times with 1 animal per group and 2 independent times, respectively, with similar results. Rabies tracing in Fig. 4 was conducted with 3 animals per group with similar results. The DREADD behavioral experiments in Fig. 5d–j were repeated 4 times independently with 3–5 animals per group with similar results.

### Reporting summary

Further information on research design is available in the Nature Portfolio Reporting Summary linked to this article.

## Data availability

All source data used to generate this manuscript are provided with this paper. Source data are provided with this paper.

## Code availability

Analysis of Z-scored delta F/F was performed in Matlab as described in[84] and readily available at https://github.com/katemartian/Photometry_data_processing.

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

## Acknowledgements

We would like to thank Mark L. Andermann, Lee B. Barrett, Nick Andrews, Yu-Ting Cheng, Mark Scimone, Jonathan M. Szeber, and David Yarmolinsky, for experimental expertise and feedback. Funding was provided by Charles Robert Broderick III Phytocannabinoid Fellowship Award (D.G.T), William Randolph Hearst Fund Fellowship (Q.J.), NIH R01EY013613 (C.C.), Tan Yang Center for Autism Research (C.C. and Q.J.), R01AT010779 (Z.H.), and BRAIN Initiative Grants funded by NCCIH/NIMH R01 AT011447 (C.J.W.) and NINDS R0NS1109947(Z.H.). In addition, we would like to thank the Harvard Neuroimaging Facility (NINDS P30 Core Center Grant #NS072030 and NIH OD 1S10OD026866), the Boston Children's Hospital Cellular Imaging Core (5P50HD105351-02), the Boston Children's Hospital Human Neuron Core (IDDRC: U54HD090255), and the Boston Children's Hospital Viral Core (NEI P30 grant: 5P30EY012196).

## Author contributions

D.G.T, C.J.W, C. Chen, and Z.H. conceived of the experiments, design, and statistical analysis, and wrote the manuscript. D.G.T. performed calcium imaging, behavioral experiments, viral tracing, and histology. F.P. and A.Carroll performed behavioral experiments and histology. Q.J. and C.Chen. performed slice electrophysiology. K.Y., C.Chung., A.Callen., performed histology and projection analysis. M.R.B. performed 2-photon serial tomography. M.E. performed RNAscope. J.S., C.G., and A.J. performed surgeries. All authors aided in manuscript editing and writing.

## Competing interests

C.J.W is a founder of Nocion Therapeutics, Quralis, and Blackbox Bio. Z.H. is a founder of Myrobalan Therapeutics and Rugen, and an advisor of Axonis. The remaining authors declare no competing interests.
