## [Peer Review File · Nature Communications]

The Secondary Somatosensory Cortex Gates Mechanical and Thermal SensitivityREVIEWER COMMENTS

Reviewer #1 (Remarks to the Author):

This manuscript is well-written and describes a novel intracortical S2-M2 circuit that modulates behavioral responsivity to mechanical and heat stimuli. Using circuit tracing and complementary optogenetic and chemogenetic approaches, the authors demonstrate the complex role of S2-to-M2 excitatory projections in gating paw withdrawal responses to mechanical and thermal stimuli. The main strengths of this study are study of a novel circuit, revealing a potentially novel mechanism (affecting motor instead of sensory perspective of the reflex), and the use of a combination of cutting-edge modern neuroscience techniques. While the manuscript would be of great interest to the field, some of the major conclusions are not adequately supported by the current data. Specific concerns are as follows:

Figure 1 major:

1) Both optogenetic (Fig 1) and chemogenetic (Fig 5) manipulations of S2 activity show reporter expression spilling over into the barrel region of S1 (see esp. Fig S1). While the authors assume that this off-target expression would not affect hindpaw responses, they do not convincingly show data or cite literature supporting this assumption. It is possible that the changes in mechanical and heat sensitivity partially resulted from unintended contributions from unknown functions of barrel region of S1. In any case, the evidence that S1 and S2 modulate distinct somatosensory modalities, as the authors propose, is not strong (only behavior data. The in vivo calcium imaging shows S2 PV+ neurons are also responsive to cold stimuli).

2) The authors use slice recordings to show that optical activation of PV neurons leads to pyramidal neuron silencing. They should show some evidence that this strategy works in vivo as anticipated, perhaps through IEG (e.g. c-Fos) immunostaining to demonstrate a reduction in activated pyramidal neurons of S2 after somatosensory stimulation during optogenetic inhibition. This could also help argue that the optical fiber locally manipulates S2 activity without more broadly affecting S1 barrel cortex activity as well.

3) The conclusion that ChR2 mice didn't show aversion to the combination of S2 inhibition and 39-degree temperature is also on the fence. The trend is there, though not significant yet, with "n" of 7, which is on the low end for behavior assays. It is possible that with higher "n" number, the P value will be "significant". Is there a reasonable sample size justification to back up this conclusion?

Minor: Fig 1J does not clearly illustrate how the authors calculated 39 degrees C as the "average" temperature at which mice withdrew their paws during PV neuron stimulation. It would be helpful to depict averaged paw withdrawal time and temperature in control and ChR2 mice.

Figure 2 major:

1) The fiber photometry data alone is difficult to interpret. The conclusion that S2 specifically suppresses responses to lower threshold, but not noxious, mechanical stimuli will need more supportive evidence. To address this, the authors could try using the optogenetics paradigm from Figure 1 at different mechanical forces to test whether the percentage of trials that result in a paw withdrawal changes when PV neurons are activated – if their model is true, inhibiting S2 output would increase the percentage of paw withdrawals for a range of non-noxious forces at the baseline condition. As inflammation shifts the "curve" of PV neuron calcium responses, so should optogenetic inhibition of S2 shift the force threshold at which light can modulate paw withdrawal behavior.

Minor:

1) The way of calculating AUC for the paw withdrawal trials is not correct in panel 2D. The baseline in

panel 2D is negative for the withdrawal curve. The AUC should be calculated to its own baseline but not the baseline of non-withdrawal curve to accurately reflect the calcium dynamics shown.

2) In Figure 2J, cooling appears to evoke a response in S2 PV neurons. It would be helpful for the authors to briefly discuss this result in the context of their conclusion that cool sensitivity is specifically influenced by S1, not S2.

Reviewer #2 (Remarks to the Author):

In the manuscript NCOMMS-23-26188-T Daniel Taub and colleagues interrogated the implication of the secondary somatosensory cortex (S2) in mechanical and thermal sensitivity by combining opto- and pharmaco-genetic approaches in the mouse model. They collected an impressive set of original data, first demonstrating an increased mechanical and heat sensitivity upon optogenetic inhibition of S2. Then, through the characterisation of long-range projections from S2, and the use of specific pharmaco-genetic manipulations, they further show that this effect is mediated through cortico-cortical projections, mainly arising from layer Va of S2 to the layer 2/3 of the secondary motor area M2. This article is likely to be of interest for a large audience by contributing to better understand how somatosensory-motor cortical circuitry governs behaviour. However, in its current state, the manuscript suffers from some limitations, and the following concerns should be addressed to allow for a better interpretation of the results.

Major concerns:

When the authors state that S2 inhibition leads to an increased mechanical and heat sensitivity, it is not immediately clear, at least to my eyes, if we are facing here an increased response to touch or an allodynic effect involving nociception related mechanisms. The authors provide experimental evidence, through a conditioned place aversion assay, that S2 inhibition, although increasing sensitivity, is not inducing aversive behaviour. But they did not test the effect of S2 inhibition on responses to non-noxious light touch, which would have contributed to disambiguate more clearly these two aspects. The absence of such assessment strongly restricts the interpretation of the results regarding the functional role of this otherwise nicely described S2-M2 connectivity.

In the abstract of the manuscript, the authors state (line 10), "we discover that S2 projections to the secondary motor cortex (M2) govern mechanical and thermal sensitivity without affecting motor or cognitive function". The word "cognitive" here is a bit strong as it covers many aspects of brain function, and I don't see from which results this assertion is coming from. I guess that they refer to the fact that they have ruled out a potential aversive or anxiogenic effect of S2 inhibition. I strongly encourage the authors to be more specific here. Regarding the motor function, in order to assess if the effect of the specific inhibition of S2 to M2 projection could be due to an altered motor reactivity, they used the sciatic functional index, and stride length quantification. Whether these locomotion related variables are really pertinent to draw strong conclusion about an eventual involvement of the pathway in motor initiation is not clear to me, this should be at least discussed.

Other concerns:

- Line 2, to my eyes, it would be more logic to say "processing and perception".
- Line 3, I suggest writing "is received primarily by two distinct regions", since it has been shown that tactile sensory information can then further flow to higher order areas (Ferezou et al., 2007, Le Merre et al., 2018 etc.).
- Line 11, there is a repetition in the sentence, "that S2" should be removed.
- Lines 30-32, it seems that the authors are describing here more a loop than a "feed-forward circuit".
- Line 37, there an extra "that", which should be removed from the sentence.
- Line 42, I'd remove here the ref 9 which doesn't strongly support a hierarchical view of S1-S2 interactions, in contrast to ref 54 which could be added here.
- Line 48, Ref 17 should come before the coma as it is a rat study.
- Line 63, if I understood well (see Fig 1b), it should be mentioned here "(ChR2) or mCherry into

parvalbumin (PV) inhibitory interneurons.”

- Lines 76, 84, 86 etc. the authors provide mean values in the main text of the manuscript, they should be accompanied by +/- standard deviations and/or P values to be more informative.
- Line 81, add “Fig 1j”. Actually, to my eyes, this panel should come closer to Fig 1g, at least before the panel i.
- Line 89, add the corresponding ref.
- Line 96, if I understood well, this temperature has been extracted given the averaged latency of withdrawal from the heat source, this later value should be indicated in the panel J of Fig 1, which should come before the panel i.
- Line 125, the authors state that there are no calcium transients in the trials in which the mice did not withdrawal their paw in the range up to 0.6g. This not so clear to my eyes at 0.6 g. Regarding the quantification of calcium signals, and the precise pre-processing steps, authors refer to a previous publication which is not open access. This is problematic, as it doesn’t allow the data to be interpreted as they should. In particular, it must be clearly indicated which reference F value is used in the delta F over F calculation. It would also be interesting to quantify averaged evoked calcium signal as a function of stimulus intensity, independently of the behavior. In direct link with this point, line 134-135, it is stated that the PV interneurons are “tuned to specific mechanical forces”, where it seems more simply that they tend to respond more for higher intensities of stimulation.
- Lines 139-141, the references to the figure panels are wrong.
- Line 150, “PV interneurons are particularly tuned to low intensity mechanical and heat stimuli” should read “PV interneurons are particularly tuned to high intensity mechanical and heat stimuli?”
- Line 167, there is an extra “now” in the sentence.
- Line 181, “quantified in Fig. 3f”, from how many animals these quantifications have been done?
- Line 228, it should be indicated that the “starter cells” are labelled through GFP expression.
- Lines 245-246 refers to the results presented in Fig. 5a et b. In fig. 5b, in S2, it would have been expected to see preferentially mCherry fluorescence in layer Va. The observed pattern of fluorescence should therefore be further commented.
- Lines 249-250, regarding the cFos experiment, the authors should assess and comment the possible increase of cFos expression in M2 following excitatory DREADD activation.
- Line 278, there is an inappropriate coma in the sentence.
- Line 283, the conclusion of this paragraph starts by “this suggest that”. However, the end of the sentence “independent of motor planning and learning” does not relate to the above-mentioned results, but to the fact that the observed altered behaviour (paw withdrawal) is likely to involve neither complex decision making nor learning.
- Line 295, the sentence should read “[...] failed to respond to heat or alter heat-evoked [...]”.
- Line 297, there is no need to introduce the S2 abbreviation again.
- Line 310, “Further” should be replaced by “Indeed”.
- Line 386, the legend indicates “in vivo calcium imaging”, this should be replaced by “in vivo calcium fiber photometry”.
- Regarding the photometry experiments, it is not clear, from the methods section, how the collected data were synchronised with the mechanical stimulations or paw withdrawal.
- Line 629, the word “have” should be removed.
- The quality of the figures in general should be improved, in particular by unifying font sizes. The text is particularly difficult to read for all the figure panels that have been generated with BioRender.
- Fig 1c, is the mark over S1-bf related to the implanted optical fiber cannula?
- Fig 1 e, it is specified that the pyramidal neuron is a representative case over 10 recorded neurons, how many PV cells have been recorded?
- Fig 1 I, k, it is really difficult to link the schematic shown in panel i with the data presented in k. Indeed, in i the behavioural chamber is composed of the two thermal plates and a central compartment, while in k it seems that the chamber is presented in a reversed manner (stripped compartment on the right) and it seems that there was no quantification of the time spent in the black compartment (on the left in k) that appears to be cropped on the image. This should be clarified to allow proper interpretation of the data presented, and a proper colorscale is missing.
- Fig 3, panel f, the legend mention S2 in red and S1 in green, but these are not on the figure. Also

panels h & i should be inverted.

- SupFig 1. How many animals are there?

Finally, the discussion section of the manuscript is rather short in its current form, the limitations of the study are scarcely discussed. The various points mentioned above, particularly with regard to my main concerns, should be discussed more deeply. Also, it would be worth mentioning the recent publication from Chang et al. (2022, <https://doi.org/10.1523/JNEUROSCI.0994-21.2021>), and take these findings in consideration while discussing the functional implication of S2 in sensorimotor integration.

Reviewer #3 (Remarks to the Author):

Drs. Taub et al. conducted experiments involving mechanical and thermal tests, both with and without the activation of PV neurons in the S2. Their findings suggest that activating PV neurons increases sensitivity to mechanical and thermal stimuli but not to cold. They argue that the pathway from the S2 to the M2 is crucial for this increased sensitivity, based on experiments using the DREADD system. While the observations related to increased sensitivity due to PV neuron activation and the role of the S2-M2 pathway might be intriguing, several concerns arise that make me question both the key findings and the subsequent interpretations. These concerns extend beyond what can be justified solely based on the available experimental evidence.

1: AAV injection to label PV-ChR2 did not restrict the expression to S2 but also spread to the S1 (and thus possibly to other regions) (Figure 1C). Although the authors specifically targeted S2 for optogenetic manipulation (Fig. 1f-l), this off-target labeling raises questions about the specificity of the observed behavioral effects. It is critical to rule out the possibility that the behavioral outcomes are influenced by the activation of S1 PV neurons at least.

2: The authors employed fiber photometry to record PV neuronal activity during mechanical and thermal stimuli, arguing that PV interneurons are more tuned to specific mechanical forces on the paw (Line 134) (Fig. 2d-g, l-o). While their interpretation suggests a causal relationship between PV activation and sensitivity, the data could equally be construed as indicating that the hindpaw movement itself evoked the increased PV activity. In Figure 2i, they attempted to address this concern by aligning the neural activity with the initiation of the movement, noting that PV activity commenced 1 second prior to hindpaw movement during the heat test. To strengthen their argument, they should apply the same technique that was used in the Fig. 2i to Fig. 2d-g. Alternatively, they could apply a generalized linear model (GLM) to statistically assess the influence of movement factors on recorded neural activity, as suggested by Musall et al. in Nature Neuroscience in 2019.

3: The comparison of AUCs between heat and cold tests (Fig. 2i-k) is problematic due to differences in stimulus modality (heat vs. acetone) (Line 145-153). A straightforward AUC comparison between these different stimulus types is questionable.

4: The DREADD system was employed with a high concentration and volume of CNO (300uM and 300 uL), much higher than cited protocols (Stachniak et al., 3 uM, 50-100uL). Such dosages could have off-target effects, affecting not only the S2-M2 pathway but also other neural circuits. The possibility of these non-specific effects should be rigorously evaluated, possibly by performing control experiments using 300uM CNO without DREADD expression in mice.

5: The authors assert that inhibition of the S2 during somatosensory stimulation leads to decreased activity in M2 (Line 235) and increased sensitivity to mechanical and thermal stimuli. Yet, they also highlight that the mouse whisker system has identified S2 as critical for decision-making (Ref 14,71) (Line 317). Furthermore, they note that projections from the S2 to the S1 are thought to be pivotal in encoding choices post-stimulation (Ref 14), and that lesions in M2 impair performance in various

sensory-based Go/No-Go tasks (Ref 41-43, 70) (Line 314). These references collectively suggest that M2 and S2 activities are important for sensory-based behaviors, which seemingly contradicts the authors' conclusions. While the authors argue that their behavioral assays are distinct because they neither involve learned behaviors nor rely on predictable stimuli (Line 321), this claim is further challenged by findings from Ref 3. That study, which also utilized passive sensory stimuli and monitored paw withdrawal, underscores the importance of M2 activity for accurate perception, thereby contradicting the authors' results. I believe the authors need to more explicitly address these inconsistencies. One plausible scenario is that M2 activity may not actually decrease in their experimental conditions but could increase through alternative mechanisms (see below). Consequently, it is imperative for the authors to closely monitor M2 activity to delineate its modulation by the S2.

6: More critically, the authors claim an increase in PV neuron activity corresponding with the rise in mechanical stimulus intensity (Fig. 2), yet they neglect to address whether activity in S2 excitatory neurons also increases. It is difficult to imagine that the increase in stimulus intensity does not correspond to an increase in Exc neuron activity. There is currently no evidence supporting the notion that increased PV activity leads to decreased S2 excitatory neuron activity (thus decreased S2 outputs) under varying stimulus intensities. As such, their assertion that mechanical stimulation inhibits S2 outputs via PV neurons is largely speculative. It would be enlightening to see data on S2 excitatory neuron activity in pre- and post-inflammation conditions. If it turns out that S2 excitatory activity does not decrease, then logically, M2 activity should also remain unchanged. Clarifying this point could help the authors resolve the apparent contradictions outlined earlier.

Minor Points:

The description of the method for monitoring paw withdrawal movement is insufficiently detailed, which is crucial for understanding how movement influences the recorded neural activity.

The introductory sentence refers to the top-down modulation of somatosensory encoding in the spinal cord (Line 24), which is misleading given that the paper does not present any data related to this process. Mention of the spinal cord (or top-down modulation by the brain) should be omitted unless relevant data are included.

REVIEWER COMMENTS

We would like to sincerely thank the reviewers for their thorough and thoughtful reviews that have strengthened the main conclusions of our paper. Major critiques arose from potential viral off-target effects, the validity of our optogenetic inhibition method, and whether inhibition in the secondary somatosensory cortex (S2) produces hypersensitivity or allodynia, among other concerns. We have addressed each point in detail below to confirm that our manipulation of S2 are specific, effective, and produce tactile allodynia rather than a generalized hypersensitivity. We address each point below in detail in blue.

Reviewer #1 (Remarks to the Author):

This manuscript is well-written and describes a novel intracortical S2-M2 circuit that modulates behavioral responsivity to mechanical and heat stimuli. Using circuit tracing and complementary optogenetic and chemogenetic approaches, the authors demonstrate the complex role of S2-to-M2 excitatory projections in gating paw withdrawal responses to mechanical and thermal stimuli. The main strengths of this study are study of a novel circuit, revealing a potentially novel mechanism (affecting motor instead of sensory perspective of the reflex), and the use of a combination of cutting-edge modern neuroscience techniques. While the manuscript would be of great interest to the field, some of the major conclusions are not adequately supported by the current data. Specific concerns are as follows:

Figure 1 major:

1) Both optogenetic (Fig 1) and chemogenetic (Fig 5) manipulations of S2 activity show reporter expression spilling over into the barrel region of S1 (see esp. Fig S1). While the authors assume that this off-target expression would not affect hindpaw responses, they do not convincingly show data or cite literature supporting this assumption. It is possible that the changes in mechanical and heat sensitivity partially resulted from unintended contributions from unknown functions of barrel region of S1. In any case, the evidence that S1 and S2 modulate distinct somatosensory modalities, as the authors propose, is not strong (only behavior data. The in vivo calcium imaging shows S2 PV+ neurons are also responsive to cold stimuli).

The somatotopic arrangement of somatosensory areas has long been shown to be faithful to representing the body map. Further, manipulation of the whisker S1 has no effect on hindpaw motor function (Chang et al., 2022 Journal of Neuroscience; Osaki et al., 2022 Nature Communications; Arakawa and Erzurumlu, 2015 Behav Brain Res; Warren et al., 2021 eLife). However, we agree that viral spill over to the barrel region needed to be formally assessed. We have now performed the same optogenetic experiments in the barrel cortex and examined von Frey mechanical thresholds and Hargreaves heat thresholds, our two major behavioral phenotypes. As shown in Supp. Fig. 2, optogenetic activation of parvalbumin inhibitory neurons in the barrel cortex of S1 (1mm adjacent medially to our coordinates in S2) had no effect on hindpaw mechanical and heat withdrawal thresholds.

We agree that our data with a cold stimulus show that PV neurons are responsive and therefore this modality is not different. However, as S1 has no heat encoding and S2 does and the effects of S1/S2 inhibition on mechanical sensitivity are in the opposite direction, we still conclude that our data (and the cited data of other labs) supports a model that there is a distinct encoding of different stimulus modalities. We have revised the manuscript to discuss differences between neural encoding and the

modulation of actions, as well as added a section in the discussion which details the shortcomings of current methods of examining cold sensitivity in rodents. See lines 363-372.

2) The authors use slice recordings to show that optical activation of PV neurons leads to pyramidal neuron silencing. They should show some evidence that this strategy works in vivo as anticipated, perhaps through IEG (e.g. c-Fos) immunostaining to demonstrate a reduction in activated pyramidal neurons of S2 after somatosensory stimulation during optogenetic inhibition. This could also help argue that the optical fiber locally manipulates S2 activity without more broadly affecting S1 barrel cortex activity as well.

We agree, and as the reviewer suggested have examined c-Fos expression in S2 after PV neuron activation. As shown in Figure 1f, optical activation of PV neurons suppresses c-Fos expression in layer V S2 pyramidal neurons, in agreement with our in vitro slice recordings from S2.

3) The conclusion that Chr2 mice didn't show aversion to the combination of S2 inhibition and 39-degree temperature is also on the fence. The trend is there, though not significant yet, with "n" of 7, which is on the low end for behavior assays. It is possible that with higher "n" number, the P value will be "significant". Is there a reasonable sample size justification to back up this conclusion?

Our lab has previously used this range of "n" for place preference assays (n of 5 per group in Michoud et al., 2021 Nature Biotechnology and n of 6 in Fell et al., 2014, Cell). However, we agree there could be a trend with the current data and have added 3 more animals to each condition. As reflected in the updated Figure 1n, any trend has disappeared with the addition of more animals and reinforces our initial conclusion that S2 optogenetic inhibition does not produce aversion.

Minor: Fig 1J does not clearly illustrate how the authors calculated 39 degrees C as the "average" temperature at which mice withdrew their paws during PV neuron stimulation. It would be helpful to depict averaged paw withdrawal time and temperature in control and Chr2 mice.

We have corrected this figure to depict when control and Chr2 mice withdrawal their paw. The average of 39 °C was captured by interpolating the temperature based on the average time of the withdrawal response for the Chr2 mice during the standard heat ramp. See Figure 1k.

Figure 2 major:

1) The fiber photometry data alone is difficult to interpret. The conclusion that S2 specifically suppresses responses to lower threshold, but not noxious, mechanical stimuli will need more supportive evidence. To address this, the authors could try using the optogenetics paradigm from Figure 1 at different mechanical forces to test whether the percentage of trials that result in a paw withdrawal changes when PV neurons are activated – if their model is true, inhibiting S2 output would increase the percentage of paw withdrawals for a range of non-noxious forces at the baseline condition. As inflammation shifts the "curve" of PV neuron calcium responses, so should optogenetic inhibition of S2 shift the force threshold at which light can modulate paw withdrawal behavior.

This is an excellent point, and we agree. We have now performed this exact experiment by optogenetically activating PV neurons and examining different low threshold and high threshold mechanical stimuli. As expected, inhibiting S2 output significantly increased the percentage of paw withdrawals for a range of non-noxious forces in Chr2 expressing animals but not in mCherry controls

(see Fig 1h). Taken together this data reinforces our conclusion that S2 suppresses responses to low intensity (innocuous) mechanical stimuli.

Minor:

1) The way of calculating AUC for the paw withdrawal trials is not correct in panel 2D. The baseline in panel 2D is negative for the withdrawal curve. The AUC should be calculated to its own baseline but not the baseline of non-withdrawal curve to accurately reflect the calcium dynamics shown.

Thank you, we have now corrected this.

2) In Figure 2J, cooling appears to evoke a response in S2 PV neurons. It would be helpful for the authors to briefly discuss this result in the context of their conclusion that cool sensitivity is specifically influenced by S1, not S2.

We have added in lines 363-372 a discussion of the differences in encoding perception vs. action modulation. S2 PV neurons do respond to cooling via acetone application with a smaller response than to heating and, their manipulation has no effect on the cooling mediated paw flinching/flicking/biting behavior.

Reviewer #2 (Remarks to the Author):

In the manuscript NCOMMS-23-26188-T Daniel Taub and colleagues interrogated the implication of the secondary somatosensory cortex (S2) in mechanical and thermal sensitivity by combining opto- and pharmaco-genetic approaches in the mouse model. They collected an impressive set of original data, first demonstrating an increased mechanical and heat sensitivity upon optogenetic inhibition of S2. Then, through the characterisation of long-range projections from S2, and the use of specific pharmaco-genetic manipulations, they further show that this effect is mediated through cortico-cortical projections, mainly arising from layer Va of S2 to the layer 2/3 of the secondary motor area M2. This article is likely to be of interest for a large audience by contributing to better understand how somatosensory-motor cortical circuitry governs behaviour. However, in its current state, the manuscript suffers from some limitations, and the following concerns should be addressed to allow for a better interpretation of the results.

Major concerns:

When the authors state that S2 inhibition leads to an increased mechanical and heat sensitivity, it is not immediately clear, at least to my eyes, if we are facing here an increased response to touch or an allodynic effect involving nociception related mechanisms. The authors provide experimental evidence, through a conditioned place aversion assay, that S2 inhibition, although increasing sensitivity, is not inducing aversive behaviour. But they did not test the effect of S2 inhibition on responses to non-noxious light touch, which would have contributed to disambiguate more clearly these two aspects. The absence of such assessment strongly restricts the interpretation of the results regarding the functional role of this otherwise nicely described S2-M2 connectivity.

We agree with this comment and have now performed the requested experiment (Figure 1h). Using our optogenetic inhibition paradigm in which PV neurons are activated with ChR2, we find that this is an allodynic effect rather than an increased response to touch overall. This is exemplified by data showing that at 0.6g-1.4g mechanical stimuli, there is no increase in the percentage of withdrawals. In contrast, the response to lower intensity stimuli are greatly enhanced.

In the abstract of the manuscript, the authors state (line 10), “we discover that S2 projections to the secondary motor cortex (M2) govern mechanical and thermal sensitivity without affecting motor or cognitive function”. The word “cognitive” here is a bit strong as it covers many aspects of brain function, and I don’t see from which results this assertion is coming from. I guess that they refer to the fact that they have ruled out a potential aversive or anxiogenic effect of S2 inhibition. I strongly encourage the authors to be more specific here. Regarding the motor function, in order to assess if the effect of the specific inhibition of S2 to M2 projection could be due to an altered motor reactivity, they used the sciatic functional index, and stride length quantification. Whether these locomotion related variables are really pertinent to draw strong conclusion about an eventual involvement of the pathway in motor initiation is not clear to me, this should be at least discussed.

We agree the wording in the abstract is too strong and non-descript and have changed the wording to specifically reflect that the anxiolytic state is unaltered.

Regarding the issue whether locomotion variables are relevant to our conclusions on motor initiation, we argue the data strongly suggests that the behavioral hypersensitivity resulting from inhibition of S2 output is not due to abnormal motor behavior and present and discuss this now in the text in Lines 317-328.

Other concerns:

- Line 2, to my eyes, it would be more logic to say “processing and perception”.

Changed.

- Line 3, I suggest writing “is received primarily by two distinct regions”, since it has been shown that tactile sensory information can then further flow to higher order areas (Ferezou et al., 2007, Le Merre et al., 2018 etc.).

Changed.

- Line 11, there is a repetition in the sentence, “that S2” should be removed.

Changed.

- Lines 30-32, it seems that the authors are describing here more a loop than a “feed-forward circuit”.

Changed wording to loop.

- Line 37, there an extra “that”, which should be removed from the sentence.

Changed.

- Line 42, I’d remove here the ref 9 which doesn’t strongly support a hierarchical view of S1-S2 interactions, in contrast to ref 54 which could be added here.

Removed Ref 9 and added Ref 54.

- Line 48, Ref 17 should come before the coma as it is a rat study.

Changed.

- Line 63, if I understood well (see Fig 1b), it should be mentioned here “(ChR2) or mCherry into parvalbumin (PV) inhibitory interneurons.”

Correct, changed to add mCherry.

- Lines 76, 84, 86 etc. the authors provide mean values in the main text of the manuscript, they should be accompanied by +/- standard deviations and/or P values to be more informative.

Standard errors are now included throughout the manuscript as requested and raw data provided in the Source Data.

- Line 81, add “Fig 1j”. Actually, to my eyes, this panel should come closer to Fig 1g, at least before the panel i.

Fig 1 was reorganized accordingly.

- Line 89, add the corresponding ref.

Reference added accordingly.

- Line 96, if I understood well, this temperature has been extracted given the averaged latency of withdrawal from the heat source, this later value should be indicated in the panel J of Fig 1, which should come before the panel i.

Changed as requested and added more detail to the graph.

- Line 125, the authors state that there are no calcium transients in the trials in which the mice did not withdraw their paw in the range up to 0.6g. This not so clear to my eyes at 0.6 g. Regarding the quantification of calcium signals, and the precise pre-processing steps, authors refer to a previous publication which is not open access. This is problematic, as it doesn't allow the data to be interpreted as they should. In particular, it must be clearly indicated which reference F value is used in the $\Delta F/F$ calculation. It would also be interesting to quantify averaged evoked calcium signal as a function of stimulus intensity, independently of the behavior. In direct link with this point, line 134-135, it is stated that the PV interneurons are "tuned to specific mechanical forces", where it seems more simply that they tend to respond more for higher intensities of stimulation.

This was an error in our wording as there is clearly a transient at 0.6g. We have modified the wording accordingly. We also expand on the precise pre-processing steps used and in addition, provide a link to the code used, as in the text below.

"Analysis of Z-scored $\Delta F/F$ was performed in Matlab as described in Martianova et al., 2019 and readily available at https://github.com/katemartian/Photometry_data_processing. In short, the raw $\Delta F/F$ is first smoothened and the slope due to fluorescent decay removed with the airPLS algorithm. The signals are then standardized and Z-scored."

- Lines 139-141, the references to the figure panels are wrong.

Corrected.

- Line 150, "PV interneurons are particularly tuned to low intensity mechanical and heat stimuli" should read "PV interneurons are particularly tuned to high intensity mechanical and heat stimuli?"

Changed accordingly.

- Line 167, there is an extra "now" in the sentence.

Changed.

- Line 181, "quantified in Fig. 3f", from how many animals these quantifications have been done?

This is indicated in the methods but now also placed in the figure legend for Fig 3f for reader clarity.

- Line 228, it should be indicated that the "starter cells" are labelled through GFP expression.

Added that they are labeled through GFP.

- Lines 245-246 refers to the results presented in Fig. 5a et b. In fig. 5b, in S2, it would have been expected to see preferentially mCherry fluorescence in layer Va. The observed pattern of fluorescence should therefore be further commented.

We have expanded on this in the figure legend. The majority of the signal is in layer Va but as you can see in Fig 3g there are also a smaller number of layer II/III and layer VI neurons that have connectivity to M2, which is why we say the signals largely originate from layer V.

- Lines 249-250, regarding the cFos experiment, the authors should assess and comment the possible increase of cFos expression in M2 following excitatory DREADD activation.

S2 to M2 neurons presumably represent only a small portion of the total inputs to the M2 region. Therefore, even though the neurons in S2 are active, cFos may be obscured in M2, especially in such a highly active pre-motor cortical region. Analysis of cFos staining in M2 would be technically challenging and unlikely to reveal a clear difference.

- Line 278, there is an inappropriate coma in the sentence.

Comma has been removed.

- Line 283, the conclusion of this paragraph starts by “this suggest that”. However, the end of the sentence “independent of motor planning and learning” does not relate to the above-mentioned results, but to the fact that the observed altered behaviour (paw withdrawal) is likely to involve neither complex decision making nor learning.

We have changed this to “This suggests that the S2 to M2 circuit is primarily sensory in nature and exerts its effects on an animals’ response to sensory stimulation independent of complex decision making nor learning.”

- Line 295, the sentence should read “[...] failed to respond to heat or alter heat-evoked [...]”.

Changed accordingly.

- Line 297, there is no need to introduce the S2 abbreviation again.

Changed back to abbreviation.

- Line 310, “Further” should be replaced by “Indeed”.

We have replaced this with “and” to connect the two sentences.

- Line 386, the legend indicates “in vivo calcium imaging”, this should be replaced by “in vivo calcium fiber photometry”.

Changed.

- Regarding the photometry experiments, it is not clear, from the methods section, how the collected data were synchronised with the mechanical stimulations or paw withdrawal.

We have now expanded on this in the methods section.

- Line 629, the word “have” should be removed.

Removed.

- The quality of the figures in general should be improved, in particular by unifying font sizes. The text is particularly difficult to read for all the figure panels that have been generated with BioRender.

We have increased font sizes for all BioRender and increased figure resolution as much as allowable.

- Fig 1c, is the mark over S1-bf related to the implanted optical fiber cannula?

This is due to tissue folding during sample processing. To avoid confusing the reader, we have now changed the representative picture in Fig 1c.

- Fig 1 e, it is specified that the pyramidal neuron is a representative case over 10 recorded neurons, how many PV cells have been recorded?

We recorded from 4 PV neurons from 4 animals and have placed this information in the Figure legend of Fig. 1.

- Fig 1 l, k, it is really difficult to link the schematic shown in panel i with the data presented in k. Indeed, in i the behavioural chamber is composed of the two thermal plates and a central compartment, while in k it seems that the chamber is presented in a reversed manner (striped compartment on the right) and it seems that there was no quantification of the time spent in the black compartment (on the left in k) that appears to be cropped on the image. This should be clarified to allow proper interpretation of the data presented, and a proper colorscale is missing.

We have changed this figure to make the thermal place preference strategy more clear. The data quantify the percentage of time spent in the optogenetic stimulus-associated chamber. Animals were conditioned in either black or striped chambers (assigned randomly) to avoid any confounding factor that the box pattern could affect preference. The likelihood that the animal is in the non-optogenetic stimulus-associated chamber is the remaining percentage, as reported in Fig 1n.

- Fig 3, panel f, the legend mention S2 in red and S1 in green, but these are not on the figure. Also panels h & i should be inverted.

This has been corrected, however, we kept h and i in the same location for ease of the reader, placing Layer II/III above Layer V.

- SupFig 1. How many animals are there?

This information has now been placed in the figure legend and in the Source Data.

Finally, the discussion section of the manuscript is rather short in its current form, the limitations of the study are scarcely discussed. The various points mentioned above, particularly with regard to my main concerns, should be discussed more deeply. Also, it would be worth mentioning the recent publication from Chang et al. (2022, <https://doi.org/10.1523/JNEUROSCI.0994-21.2021>), and take these findings in consideration while discussing the functional implication of S2 in sensorimotor integration.

We have now expanded the discussion and include Chang et al., 2022 into our discussion of sensorimotor integration.

Reviewer #3 (Remarks to the Author):

Drs. Taub et al. conducted experiments involving mechanical and thermal tests, both with and without the activation of PV neurons in the S2. Their findings suggest that activating PV neurons increases sensitivity to mechanical and thermal stimuli but not to cold. They argue that the pathway from the S2 to the M2 is crucial for this increased sensitivity, based on experiments using the DREADD system. While the observations related to increased sensitivity due to PV neuron activation and the role of the S2-M2 pathway might be intriguing, several concerns arise that make me question both the key findings and the subsequent interpretations. These concerns extend beyond what can be justified solely based on the available experimental evidence.

1: AAV injection to label PV-ChR2 did not restrict the expression to S2 but also spread to the S1 (and thus possibly to other regions) (Figure 1C). Although the authors specifically targeted S2 for optogenetic manipulation (Fig. 1f-l), this off-target labeling raises questions about the specificity of the observed behavioral effects. It is critical to rule out the possibility that the behavioral outcomes are influenced by the activation of S1 PV neurons at least.

We agree this experiment is crucial for the proper interpretation of our data, a concern also shared by Reviewer 1. We have addressed this above but paste our comment here for convenience: "The somatotopic arrangement of the somatosensory areas has long been shown to be faithful to representing the body map. Further, manipulation of the whisker S1 for example, has been shown to have no effect on hindpaw motor function (Chang et al., 2022 Journal of Neuroscience; Osaki et al., 2022 Nature Communications; Arakawa and Erzurumlu, 2015 Behav Brain Res; Warren et al., 2021 eLife). However, we agree that the viral spill over to the barrel region needed to be formally assessed. We have now performed the same optogenetic experiments in the barrel cortex and examined von Frey mechanical thresholds and Hargreaves heat thresholds, our two major behavioral phenotypes. As shown in Supp. Fig. 2, optogenetic activation of parvalbumin inhibitory neurons in the barrel cortex of S1 (1mm adjacent medially to our coordinates in S2) had no effect on mechanical and heat withdrawal thresholds."

2: The authors employed fiber photometry to record PV neuronal activity during mechanical and thermal stimuli, arguing that PV interneurons are more tuned to specific mechanical forces on the paw (Line 134) (Fig. 2d-g, l-o). While their interpretation suggests a causal relationship between PV activation and sensitivity, the data could equally be construed as indicating that the hindpaw movement itself evoked the increased PV activity. In Figure 2i, they attempted to address this concern by aligning the neural activity with the initiation of the movement, noting that PV activity commenced 1 second prior to hindpaw movement during the heat test. To strengthen their argument, they should apply the same technique that was used in the Fig. 2i to Fig. 2d-g. Alternatively, they could apply a generalized linear model (GLM) to statistically assess the influence of movement factors on recorded neural activity, as suggested by Musall et al. in Nature Neuroscience in 2019.

Inspired by the approach taken by Musall et al., 2019 Nature Neuroscience, we have now examined the fiber photometry signals when a mouse makes a non-stimulus induced hindpaw withdrawal. Given the somatotopy of the somatosensory cortices, this approach is more suited to deal with this question rather than examining bulk movement. As shown in Supp. Fig. 4a, a non-stimulus induced paw withdrawal does not produce a robust calcium transient, suggesting that sensory input is crucial for the signal and that hindpaw movement is not responsible for the evoked increase in PV activity. This is further exemplified by the data at high mechanical thresholds where in the absence of a motor response a calcium transient is still evoked. We therefore conclude this is a sensory driven response.

3: The comparison of AUCs between heat and cold tests (Fig. 2i-k) is problematic due to differences in stimulus modality (heat vs. acetone) (Line 145-153). A straightforward AUC comparison between these different stimulus types is questionable.

We agree and did not compare them statistically in the text, but did present them on the same graph. We have now separated them see Fig.2i and 2j.

4: The DREADD system was employed with a high concentration and volume of CNO (300uM and 300 uL), much higher than cited protocols (Stachniak et al., 3 uM, 50-100uL). Such dosages could have off-target effects, affecting not only the S2-M2 pathway but also other neural circuits. The possibility of these non-specific effects should be rigorously evaluated, possibly by performing control experiments using 300uM CNO without DREADD expression in mice.

We used 300 nanoliters of solution rather than 300 microliters, which is in line with the upper limits used by Stachniak et al., 2014 (see Figure 3e). The secondary motor cortex (M2) is also much larger in area than the brain region targeted by Stachniak (PVH of the hypothalamus). However, we agree that the dose is at the higher end and have now evaluated the possibility of off target effects by injecting 300 nanoliters of 300 micromolar CNO into mice without DREADD expression. As shown in Supplementary Figure 7, injecting 300 nanoliters of 300 micromolar CNO in control mice produced no effect on mechanical or heat thresholds.

5: The authors assert that inhibition of the S2 during somatosensory stimulation leads to decreased activity in M2 (Line 235) and increased sensitivity to mechanical and thermal stimuli. Yet, they also highlight that the mouse whisker system has identified S2 as critical for decision-making (Ref 14,71) (Line 317). Furthermore, they note that projections from the S2 to the S1 are thought to be pivotal in

encoding choices post-stimulation (Ref 14), and that lesions in M2 impair performance in various sensory-based Go/No-Go tasks (Ref 41-43, 70) (Line 314). These references collectively suggest that M2 and S2 activities are important for sensory-based behaviors, which seemingly contradicts the authors' conclusions. While the authors argue that their behavioral assays are distinct because they neither involve learned behaviors nor rely on predictable stimuli (Line 321), this claim is further challenged by findings from Ref 3. That study, which also utilized passive sensory stimuli and monitored paw withdrawal, underscores the importance of M2 activity for accurate perception, thereby contradicting the authors' results. I believe the authors need to more explicitly address these inconsistencies. One plausible scenario is that M2 activity may not actually decrease in their experimental conditions but could increase through alternative mechanisms (see below). Consequently, it is imperative for the authors to closely monitor M2 activity to delineate its modulation by the S2.

We agree that we need to be more explicit regarding the roles of S2/M2 in perception of vs. the motor output to a stimulus in the context of the existing literature and have changed the text accordingly. Our data show that paw withdrawal reflexes can be modulated in the absence of a learned paradigm by a cortico-cortical circuit.

We also agree that the effects on M2 are complex given the existing data in the literature and have tapered our discussion accordingly. It will be important for future studies to use a combination of circuit manipulation of sensory regions combined with single cell imaging in M2 combined with either specific Cre-drivers or post-hoc alignment of neural recordings with spatial transcriptomics to reveal which cell populations are affected, how, and what the consequences are, which is though out of the scope of the current study which focuses on S2 rather than M2.

6: More critically, the authors claim an increase in PV neuron activity corresponding with the rise in mechanical stimulus intensity (Fig. 2), yet they neglect to address whether activity in S2 excitatory neurons also increases. It is difficult to imagine that the increase in stimulus intensity does not correspond to an increase in Exc neuron activity. There is currently no evidence supporting the notion that increased PV activity leads to decreased S2 excitatory neuron activity (thus decreased S2 outputs) under varying stimulus intensities. As such, their assertion that mechanical stimulation inhibits S2 outputs via PV neurons is largely speculative. It would be enlightening to see data on S2 excitatory neuron activity in pre- and post-inflammation conditions. If it turns out that S2 excitatory activity does not decrease, then logically, M2 activity should also remain unchanged. Clarifying this point could help the authors resolve the apparent contradictions outlined earlier.

We chose to examine PV neurons because of their broad tuning properties, in contrast to excitatory neurons which tend to be more highly tuned with a scattered architecture, which we anticipated would render fiber photometry signals difficult to interpret. However, we have now performed excitatory neuron fiber photometry both pre and post-inflammation, as requested, to assess this point, and reveal that the dynamics of the excitatory neurons in S2 are indeed variable. Two specific data points address the reviewer's questions. 1. Inducing peripheral inflammation, which significantly enhances the response of PV neurons, is concomitant with a lack of a calcium response at 0.6g in the excitatory neurons. 2. The Z-scored activity in our heat ramp assay, demonstrates that overall excitatory neuron activity is low when PV neuron activity is high. Future analysis at the single cell level will be required to delineate this point more conclusively.

Minor Points:

The description of the method for monitoring paw withdrawal movement is insufficiently detailed, which is crucial for understanding how movement influences the recorded neural activity.

We expand on this in the Methods section. For each behavioral assay, we provide information on exactly how withdrawal was monitored.

The introductory sentence refers to the top-down modulation of somatosensory encoding in the spinal cord (Line 24), which is misleading given that the paper does not present any data related to this process. Mention of the spinal cord (or top-down modulation by the brain) should be omitted unless relevant data are included.

This introduction allows the reader to understand the context of our lab's work on the sensory cortex, the motivation for examining the current question, and also presents the novelty of our findings showing that a different cortical region projecting to a region other than the spinal cord impacts a behavioral response to a sensory stimulus. We would prefer to keep this introduction but can take it out if the reviewer prefers.

REVIEWERS' COMMENTS

Reviewer #1 (Remarks to the Author):

The authors have addressed the concerns raised in the last round. There are two minor points:

1) Line 110, the authors called the increase of tactile sensitivity as allodynia (low-threshold stimuli now being perceived as noxious). However, there is no evidence to support that these paw withdrawals at baseline condition indicating nociception. It seems that "hypersensitivity" is more accurate here.

2) Line 296, redundancy of "labelling" and "labeled" and a typo "thorough".

Reviewer #2 (Remarks to the Author):

Overall the authors have made a valuable effort to address all the points raised on the first version of the manuscript. The manuscript has been globally improved and deserves to be presented to the scientific community.

Few minor comments on this revised version of the manuscript:

- Line 54 should read "This constitutes a loop [...]"
- New supplementary figure 2: the panel a) should be presented as fig 1c with indications of area borders and cannula position. The authors indicate, regarding the results presented in panel b), "Light stimulation of mice with ChR2 expression in S1-barrel cortex display no tactile hypersensitivity by von Frey. n=5 mCherry, n=6 ChR2", while to my eyes it's not so clear that ChR2 stimulation has no effect. Which statistical test has been used to support this assertion? Please indicate the test and the P value.
- New figure 1f: pictures of stained cortical sections could be useful (could be provided as supplementary material), and a few more details about the counting procedure should be provided.
- On the pdf version of the manuscript I could download, the graphical quality of figure 1b and 1l is really not optimal and should be improved.

Reviewer #3 (Remarks to the Author):

The authors have addressed my concerns in the revised manuscript, and I recommend it for publication in Nature Communications.

We would like to thank the reviewers again for their detailed reading and critique of our manuscript. We address each comment below in blue.

REVIEWERS' COMMENTS

Reviewer #1 (Remarks to the Author):

The authors have addressed the concerns raised in the last round. There are two minor points:

1) Line 110, the authors called the increase of tactile sensitivity as allodynia (low-threshold stimuli now being perceived as noxious). However, there is no evidence to support that these paw withdrawals at baseline condition indicating nociception. It seems that "hypersensitivity" is more accurate here.

We believe allodynia is the correct term for what we are observing as hypersensitivity would imply that reactivity toward all forces of mechanical stimuli would be enhanced. In contrast, we observe only low threshold stimuli reactivity being enhanced which is allodynia.

2) Line 296, redundancy of "labelling" and "labeled" and a typo "thorough".

Thank you, we have now corrected this.

Reviewer #2 (Remarks to the Author):

Overall the authors have made a valuable effort to address all the points raised on the first version of the manuscript. The manuscript has been globally improved and deserves to be presented to the scientific community.

Few minor comments on this revised version of the manuscript:

- Line 54 should read "This constitutes a loop [...]"

Thank you, we have now corrected this.

- New supplementary figure 2: the panel a) should be presented as fig 1c with indications of area borders and cannula position. The authors indicate, regarding the results presented in panel b), "Light stimulation of mice with ChR2 expression in S1-barrel cortex display no tactile hypersensitivity by von Frey. n=5 mCherry, n=6 ChR2", while to my eyes it's not so clear that ChR2 stimulation has no effect. Which statistical test has been used to support this assertion? Please indicate the test and the P value.

We have now added areas of borders and cannula position to this image as requested. We used a Two-Way ANOVA followed by Sidaks posthoc tests. We have now placed p-values directly into the figure legend.

- New figure 1f: pictures of stained cortical sections could be useful (could be provided as supplementary material), and a few more details about the counting procedure should be provided.

We have now added these pictures to Supp. Fig. 1 as requested and we have now provided more details on the procedure beginning at line 661: "Staining proceeded as detailed above. The S2 region was then imaged with a Leica SP8 Confocal Microscope. Cells positive for c-fos within

layer V of the area of viral injection (as established by expression of tdTomato) were quantified and compared statistically.”

- On the pdf version of the manuscript I could download, the graphical quality of figure 1b and 11 is really not optimal and should be improved.

We have improved these.

Reviewer #3 (Remarks to the Author):

The authors have addressed my concerns in the revised manuscript, and I recommend it for publication in Nature Communications.